# A prebiotic basis for ATP as the universal energy currency

**Silvana Pinna**[1], **Cäcilia Kunz**[1], **Aaron Halpern**[1], **Stuart A. Harrison**[1], **Sean F. Jordan**[1], **John Ward**[2], **Finn Werner**[3], **Nick Lane**[1]*

**1** Centre for Life's Origins and Evolution (CLOE), Department of Genetics, Evolution and Environment, University College London, Darwin Building, London, United Kingdom, **2** Department of Biochemical Engineering, University College London, London, United Kingdom, **3** Institute for Structural and Molecular Biology, University College London, Darwin Building, London, United Kingdom

* nick.lane@ucl.ac.uk

**Data Availability Statement:** All relevant data are within the paper and its Supporting Information files.

**Funding:** We are grateful to the Biotechnology and Biological Sciences Research Council to NL, FW

## Abstract

ATP is universally conserved as the principal energy currency in cells, driving metabolism through phosphorylation and condensation reactions. Such deep conservation suggests that ATP arose at an early stage of biochemical evolution. Yet purine synthesis requires 6 phosphorylation steps linked to ATP hydrolysis. This autocatalytic requirement for ATP to synthesize ATP implies the need for an earlier prebiotic ATP equivalent, which could drive protometabolism before purine synthesis. Why this early phosphorylating agent was replaced, and specifically with ATP rather than other nucleoside triphosphates, remains a mystery. Here, we show that the deep conservation of ATP might reflect its prebiotic chemistry in relation to another universally conserved intermediate, acetyl phosphate (AcP), which bridges between thioester and phosphate metabolism by linking acetyl CoA to the substrate-level phosphorylation of ADP. We confirm earlier results showing that AcP can phosphorylate ADP to ATP at nearly 20% yield in water in the presence of $Fe^{3+}$ ions. We then show that $Fe^{3+}$ and AcP are surprisingly favoured. A wide range of prebiotically relevant ions and minerals failed to catalyse ADP phosphorylation. From a panel of prebiotic phosphorylating agents, only AcP, and to a lesser extent carbamoyl phosphate, showed any significant phosphorylating potential. Critically, AcP did not phosphorylate any other nucleoside diphosphate. We use these data, reaction kinetics, and molecular dynamic simulations to infer a possible mechanism. Our findings might suggest that the reason ATP is universally conserved across life is that its formation is chemically favoured in aqueous solution under mild prebiotic conditions.

## Introduction

ATP is casually referred to as the "universal energy currency" of life. Why it gained this ascendency in metabolism, in place of many possible equivalents, is an abiding mystery in biology. There is nothing particularly special about the "high-energy" phosphoanhydride bonds in ATP. Rather, its ability to drive phosphorylation or condensation reactions reflects the

and JW (BB/V003542/1; https://www.ukri.org/councils/bbsrc/) and HR (LIDo Doctoral Training Programme; https://www.lido-dtp.ac.uk/), to Gates Ventures (formerly bgc3) to NL, and to the Natural Environment Research Council to AH and NL (2236041; https://www.ukri.org/councils/nerc/). The funders had no role in study design, data collection and analysis, decision to publish, or preparation of the manuscript.

**Competing interests:** The authors have declared that no competing interests exist.

**Abbreviations:** AcP, acetyl phosphate; CDP, cytidine diphosphate; CP, carbamoyl phosphate; cTMP, cyclic trimetaphosphate; DAP, diamidophosphate; GDP, guanosine diphosphate; IDP, inosine diphosphate; MD, molecular dynamic; PEP, phosphoenolpyruvate; PPi(III), pyrophosphite; PPi(V), pyrophosphate; NDP, nucleoside diphosphate; TMP, trimethyl phosphate; UDP, uridine diphosphate.

extraordinary disequilibrium between ATP and ADP—about 10 orders of magnitude in modern cells, pushed by free energy derived from respiration [1]. ATP drives intermediary metabolism through the coupling of exergonic to endergonic reactions via phosphorylation and hydrolysis, but other phosphorylating agents could be pushed equally far from equilibrium and accomplish equivalent coupling.

A partial explanation is that ATP links energy metabolism with genetic information [2]. ATP-coupled monomer activation promotes the polymerisation of macromolecules, including RNA, DNA, and proteins. Unlike the simple phosphorylation of intermediary metabolites, the leaving group during nucleotide polymerization is pyrophosphate (PPi) [3]. Likewise, the activation of amino acids by adenylation liberates PPi as the leaving group [4–7]. The hydrolysis of PPi renders these steps strongly exothermic, if not practically irreversible—a ratchet towards polymerization [3,8,9]. Only nucleoside triphosphates can release PPi while still retaining a phosphate for the sugar-phosphate backbone of RNA and DNA, or for amino acid activation. But the fact that the canonical nucleosides can all form triphosphates, with equivalent free-energy profiles, only serves to reemphasise the prominence of ATP over GTP, TTP, UTP, or CTP in RNA, DNA, and protein synthesis. While GTP is not uncommon in metabolic processes, including ribosomal GTPases [10], it hardly displaces ATP from its preeminent position in biology.

In our view, the prominence of ATP is unlikely to reflect a frozen accident, as adenine nucleotide cofactors are also ubiquitous across intermediary metabolism, including the ancient cofactors NADH, FADH, and coenzyme A. The centrality of these cofactors to intermediary metabolism, combined with their ability to catalyse the same reactions in the absence of enzymes [11–14], suggests that adenosine arose very early in biology, possibly even in a monomer world before the advent of RNA, DNA, and proteins [15,16]. This hypothesis is consistent with the idea that life acts as a guide to its own origin [17,18]. Phylogenetics indicates that the earliest cells grew autotrophically from $H_2$ and $CO_2$ [16,19–23]. Recent experimental work shows that the core of autotrophic metabolism can occur spontaneously in the absence of genes and enzymes. This includes nonenzymatic equivalents of the acetyl CoA pathway and parts of the reverse Krebs cycle [24–26], glycolysis and the pentose phosphate pathway [27,28], gluconeogenesis [29], and amino acid biosynthesis [30–32]. Recent work demonstrates that some nucleobases can also be formed following the universally conserved biosynthetic pathways, using transition metal ions as catalysts [33]. The idea that ATP could have arisen as a product of protometabolism starting from $H_2$ and $CO_2$ is therefore not unreasonable (though we do not dispute that other mechanisms could potentially also give rise to purine nucleotides; [34,35]). Nonetheless, biological purine synthesis specifically involves 6 phosphorylation steps that are catalysed by ATP in modern cells—an autocatalytic feedback loop. If ATP was indeed formed in a monomer word via a biomimetic protometabolism, then an earlier ATP equivalent must have driven the phosphorylation steps in purine synthesis. Why this earlier phosphorylating agent was replaced, and specifically with ATP rather than other nucleoside triphosphates, might explain why ATP later rose to prominence in metabolism.

Arguably the most plausible ancestral mechanism of ATP synthesis is through the substrate-level phosphorylation of ADP to ATP by acetyl phosphate (AcP), which is still the fulcrum between thioester and phosphate metabolism in bacteria and archaea [36,37]. In modern bacteria, AcP is formed by the phosphorolysis of acetyl CoA; in archaea and eukaryotes, AcP remains bound to the active site of the enzyme but is still formed as a transient intermediate [37]. The notion that AcP played an important role at the origin of life goes back to Lipmann [38] and has been advocated by de Duve, Ferry and House, Martin and Russell, and others [23,36,37,39–41]. While CoA itself is derived from ATP, simpler thioesters, with equivalent functional chemistry to acetyl CoA, have long been linked with prebiotic chemistry and the core metabolic networks in cells [36,38,42–49]. Recent work suggests that thioesters such as

methyl thioacetate can be synthesised under hydrothermal conditions [50]. AcP can also be made in water under ambient or mild hydrothermal conditions by phosphorolysis of thioacetate, which, as a thiocarboxylic acid, is even simpler than thioesters [15].

AcP will phosphorylate various nucleotide precursors in water, including ribose to ribose-5-phosphate, and adenosine to AMP [15]. Importantly, AcP can also phosphorylate ADP to ATP at approximately 20% yield in water in the presence of $Fe^{3+}$ ions, suggesting that substrate-level phosphorylation could indeed take place in aqueous prebiotic conditions [51,52]. But whether this serendipitous discovery holds real relevance to protometabolism is uncertain, as other metal ions, phosphorylating agents and nucleoside diphosphates have not been tested under equivalent conditions. We have therefore explored the phosphorylation of ADP more systematically using a range of prebiotically plausible and biologically relevant phosphorylating agents, and a panel of metal ions as possible catalysts. We find that the combination of $Fe^{3+}$ and AcP is unique: No other metal ions or phosphorylating agents are as effective at phosphorylating ADP. Equally striking, we find that ADP is also unique: The combination of AcP and $Fe^{3+}$ will phosphorylate ADP but not GDP, CDP, UDP, or IDP. We use these data, reaction kinetics, and molecular dynamic (MD) simulations to propose a possible mechanism. Our results suggest that ATP became established as the universal energy currency in a prebiotic, monomeric world, on the basis of its unusual chemistry in water.

## Results

### $Fe^{3+}$ is unique in promoting ADP phosphorylation by acetyl phosphate

We analysed a panel of metal ions commonly used as cofactors in metabolism, and likely available at the origin of life, to compare their effect on the phosphorylation of ADP by AcP. We first confirmed the results of Kitani and colleagues [51,52] in demonstrating that $Fe^{3+}$ catalyses the formation of ATP by AcP at approximately 15% to 20% yield depending on the conditions (**Fig 1A**). We corroborated our HPLC results using MS/MS (**Fig 1C**) and established that $Fe^{3+}$ was not catalysing the disproportionation reaction between 2 ADP molecules (**Fig A in S1 Results**). Surprisingly, we found that $Fe^{3+}$ is uniquely effective at catalysing ADP phosphorylation, at least among the large panel of metal ions we tested. FeS clusters chelated by monomeric cysteine initially seemed to produce small yields of ATP, as shown in **Fig 1B**. However, Cys-FeS clusters are unstable and break down over hours except under strictly anoxic conditions [53]. We therefore suspected that the ATP yield actually reflected the release of $Fe^{3+}$ into the medium. This was confirmed under more strictly anoxic conditions in an anaerobic glovebox, wherein FeS clusters failed to catalyse ATP formation (**Fig B in S1 Results**).

Metal ions that are commonly associated with ATP in metabolism, notably $Mg^{2+}$ [54,55], failed to catalyse ATP formation either as free ions, or when coordinated by the monomeric amino acid aspartate, or in mineral form as brucite (**Fig C in S1 Results**). We had anticipated that chelated metal ions would show a stronger catalytic efficacy than free ions, as the coordination environment partially mimics the active site of enzymes, in this case acetate kinase or RNA polymerase (where glutamate or aspartate chelates $Mg^{2+}$ at the active site). Brucite is a hydroxide mineral ($Mg(OH)_2$) with a unit-cell structure that is also reminiscent of the $Mg^{2+}$ coordination by the carboxylate of aspartate in the RNA polymerase. Surface catalysis may play an important role in prebiotic chemistry, but in this case failed to promote ATP synthesis. $Mn^{2+}$, which has a similar activity to $Mg^{2+}$ in acetate kinase [56] also failed to promote ATP synthesis.

### ADP phosphorylation occurs in a range of aqueous prebiotic environments

We next explored the conditions under which $Fe^{3+}$ catalyses the phosphorylation of ADP by AcP, specifically pH, temperature, water activity, and pressure. We found that the reaction is

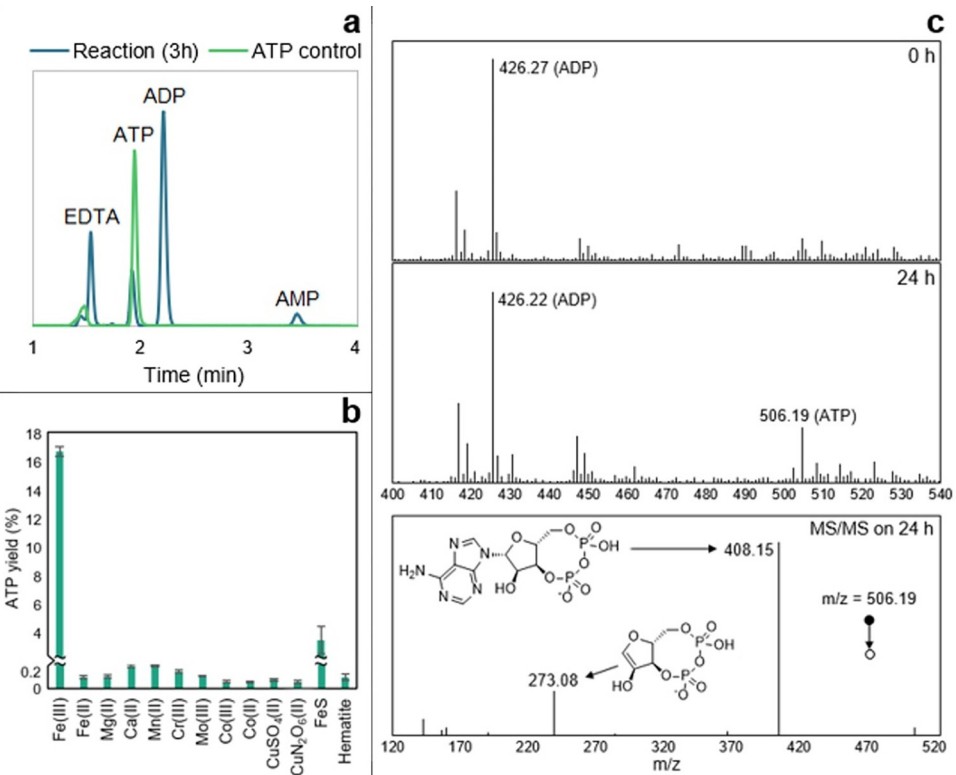

**Fig 1. ATP synthesis with metal ion catalysts.** (**a**) HPLC trace of ATP control (0.7 mM) and ATP produced by the reaction ADP (1 mM) + AcP (4 mM) + $Fe^{3+}$ (500 μM) at 30°C and pH approximately 5.5–6. (**b**) Test of reaction ADP (1 mM) + AcP (4 mM) at 30°C and pH approximately 5.5–6 with $Fe^{3+}$ ($Fe_2(SO_4)_3$), $Mg^{2+}$ ($MgCl_2$), $Ca^{2+}$ ($CaCl_2$), $Mn^{2+}$ ($Mn(NO_3)_2$), $Cr^{3+}$ ($Cr(NO_3)_3$), $Mo^{3+}$ ($MoCl_3$), $Co^{3+}$ ($[Co(NH_3)_6]Cl_3$), $Co^{2+}$ ($CoCl_2$), $CuSO_4$, $Cu(NO_3)_2$, FeS clusters (500 μM), and hematite ($Fe_2O_3$, 50 mg). The bars represent the ATP yield after 5 h. $N = 3$ ±SD. (**c**) Mass spectrometry analysis on a reaction sample at t = 0 h (upper panel) and 24 h (middle panel). The MS/MS spectrum and proposed structures of the products of the fragmentation of the ATP mass detected at 24 h (m/z = 506.19) is shown in the lower panel and was confronted to commercial standards and public data [137]. Conditions: ADP (1 mM) + AcP (4 mM) + $Fe^{3+}$ (500 μM) at 30°C and pH approximately 5.5–6. The data underlying this figure can be found in Table A–C in S1 Data (sheet 1).

strongly sensitive to pH, and occurs most readily under mildly acidic conditions, with an optimum pH of approximately 5.5 to 6, the uncorrected default pH of the reaction (**Fig 2A**). Slightly more acidic conditions (pH 4) suppressed the yield a little, but more alkaline conditions had a much stronger suppressive effect. ATP yield fell by around three-quarters at pH 7 and collapsed to nearly zero at pH 9. This collapse of phosphorylation under alkaline conditions most likely reflected the precipitation of the catalyst as $Fe(OH)_3$. While this sharp sensitivity to pH might seem at first sight limiting, in the Discussion, we show that, on the contrary, it could be valuable in generating disequilibria, enabling ATP hydrolysis to power work.

ATP yield was less acutely sensitive to temperature, at least between 20 and 50°C. Over 24 h, the overall ATP yield reflects both synthesis and hydrolysis. We found that 30°C optimised yield across 24 h, by promoting synthesis within the first 4 h while limiting hydrolysis over the subsequent 20 h (**Fig 2B**). The rate of synthesis was a little lower at 20°C, but this was offset by slightly less hydrolysis over 24 h. ATP synthesis was markedly faster at 50°C, but so too was hydrolysis, which already lowered yields within the first 2 h and cut them to less than a quarter of those at 30°C after 24 h. If ATP is to power work, as in modern cells, then hydrolysis in itself is not an issue, but rather needs to be coupled to other reactions such as the phosphorylation

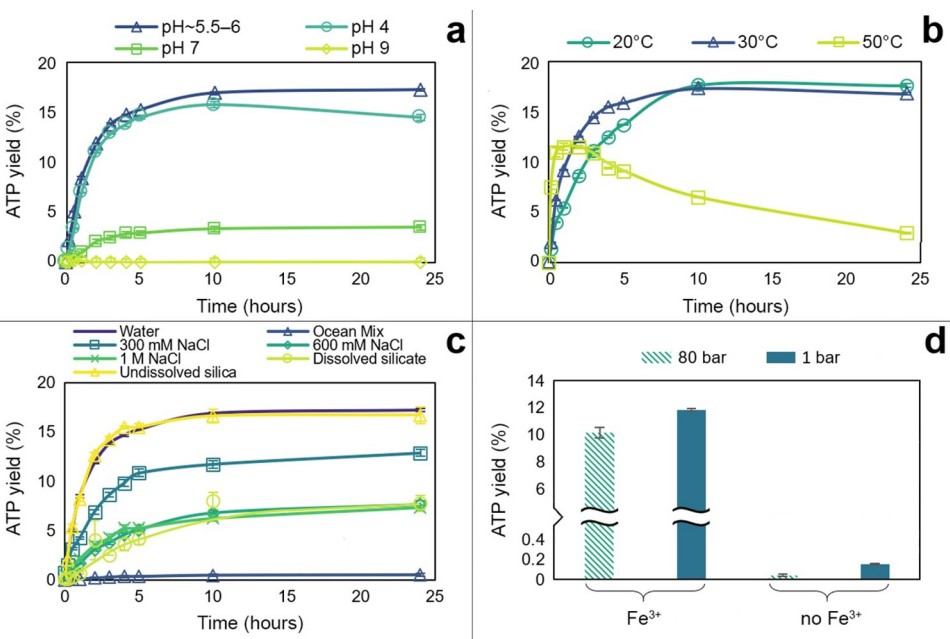

**Fig 2. ATP synthesis by AcP and $Fe^{3+}$ at different conditions.** (**a**) Effect of pH on reaction ADP (1 mM) + AcP (4 mM) + $Fe^{3+}$ (500 μM) at 30˚C. The optimal pH of the reaction is approximately 5.5–6. Rate of reaction: 0.0079 μM/s (optimal pH), 0.0074 μM/s (pH 4), and 0.0011 μM/s (pH 7). $N$ = 3 ±SD. (**b**) Effect of temperature on reaction ADP (1 mM) + AcP (4 mM) + $Fe^{3+}$ (500 μM), pH approximately 5.5–6. Rate of reaction: 0.0066 μM/s (20˚C), 0.0079 μM/s (30˚C), and 0.028 μM/s (50˚C). $N$ = 3 ±SD. (**c**) Comparison of ATP yield from the reaction ADP (1 mM) + AcP (4 mM) at 30˚C, pH approximately 5.5–6 in water (reaction ionic strength = 3.75 mM), a modern ocean mix (600 mM NaCl, 50 mM $MgCl_2$, and 10 mM $CaCl_2$, reaction ionic strength = 783.75 mM), 300 mM NaCl (reaction ionic strength = 303.75 mM), modern ocean concentration of NaCl (600 mM, reaction ionic strength = 603.75 mM), 1 mM NaCl (reaction ionic strength = 1.004 M), dissolved silicate (10 mM $SiO_2$, reaction ionic strength = 123.75 mM), and suspended silica in water (50 mg). $N$ = 3 ±SD. (**d**) Comparison of ATP yield from the reaction ADP (1 mM) + AcP (4 mM) at 30˚C and pH approximately 5.5–6 with and without $Fe^{3+}$ (500 μM) at 80 bar (striped yellow) and at atmospheric pressure (1 bar, solid blue). $N$ = 2 ±SD. The data underlying this figure can be found in Table D–G in S1 Data (sheet 2).

or condensation of substrates. Such processes also tend to take place over minutes to hours [15], meaning that temperature has a relatively trivial effect, with the yield after 2 to 3 h being similar at all 3 temperatures studied, at around 10% to 15% (**Fig 2B**). This implies that temperature would not be a strong limiting factor on many possible prebiotic environments.

More surprisingly, ATP yield was greatest at high water activity, either in HPLC-grade water or in suspended silica (**Fig 2C**). Adding NaCl lowered ATP yield, albeit not dramatically. Moderate NaCl concentration (300 mM, giving a total reaction ionic strength of 303.75 mM) lowered ATP yield by around a fifth. Modern ocean salinity (600 mM NaCl, reaction ionic strength 603.75 mM) and higher salinity (1 M NaCl, reaction ionic strength 1.004 M) both roughly halved the yield. This suggests that the effect of solutes does not only reflect ionic strength, which was confirmed by the addition of other solutes. Dissolved silicate (10 mM $SiO_2$) also halved ATP yield, even though the ionic strength in this case was only 123.75 mM (**Fig 2C**). Likewise, higher $Mg^{2+}$ and $Ca^{2+}$ concentrations (50 mM and 10 mM, respectively) as part of a modern ocean mix collapsed ATP yields to nearly zero (**Fig 2C**), presumably because $Ca^{2+}$ and $Mg^{2+}$ promote ATP hydrolysis [57,58]. While this might suggest that ATP synthesis could not occur in modern oceans, $Mg^{2+}$ and $Ca^{2+}$ concentrations can in fact vary considerably in ocean environments (see Discussion). We show later that lower $Mg^{2+}$ and $Ca^{2+}$ concentrations (approximately 2 mM) actually promote ATP synthesis.

High pressure (80 bar) had very little effect on ATP synthesis (**Fig 2D**). This is consistent with the work of Leibrock and colleagues [59], who showed that high pressure promotes ATP hydrolysis, but only at pressures $\geq$300 bar. The slightly greater ATP yield at ambient pressure in our experiment may be attributable to greater evaporation in the open (nonpressurized) system. This was clearly the case in the absence of $Fe^{3+}$, where most of the ATP detected was not produced by phosphorylation of ADP, but contamination of the ADP commercial standard via the manufacturing process, then concentrated by evaporation at ambient pressure (**Fig D in S1 Results**).

## Acetyl phosphate is more effective than other prebiotic phosphorylating agents

We compared AcP with a panel of 6 other potentially prebiotic phosphorylating agents, including a number still used by cells today (**Table 1**).

Given the diverse reaction kinetics anticipated with these different phosphorylating agents, we carried out experiments at both at 30°C (the optimal temperature for AcP) and 50°C (as most phosphate donors are less labile than AcP and so might be more effective at higher temperatures), as well as pH 5.5 to 6, 7, and 9. As shown in **Fig 3**, no other phosphorylating agent was as effective as AcP at synthesising ATP in the presence of $Fe^{3+}$. The only other phosphorylating agent to show any notable efficacy was carbamoyl phosphate (CP), which is similar in structure to AcP; it has a carbamate ($-CO-NH_2$) rather than acetate ($-CO-CH_3$) bound to phosphate. CP produced about half the ATP yield of AcP at 20°C and pH 5.5 to 6 (**Fig 3A**), but barely a quarter of the yield at pH 7 (**Fig 3B**). At pH 9, only cyclic trimetaphosphate (cTMP) produced any ATP at all, albeit after a delay of more than 20 h (**Fig 3C**).

At 50°C, CP generated ATP continuously over 24 h at pH 5.5 to 6, despite producing only half the yield in the first 2 h (**Fig 3D**). The fact that ATP yield declined over time with AcP indicates that ATP was hydrolysed over hours at 50°C; it was not replenished because AcP also hydrolysed at that temperature [15]. While CP has a similarly low thermal stability, the primary decomposition product is cyanate [73], which is itself a proficient condensing agent [74,75]. This likely contributed to a balance between the synthesis and hydrolysis of ATP over 24 h. Only AcP formed any ATP at 50°C and pH 7 (**Fig 3E**), consistent with the pH sensitivity of CP seen at 30°C. The main conclusion here is that from a panel of 7 plausibly prebiotic phosphorylating agents, only AcP was capable of generating an ATP yield of >10% in water at both 30 and 50°C. The only other agent to show remotely comparable efficacy at mildly acidic pH was CP, but its maximal yield was half that of AcP.

**Table 1. Phosphorylating agents tested.**

| Name | ID | Formula | Prebiotic/biochemical prominence |
|---|---|---|---|
| Cyclic trimetaphosphate | cTMP | $Na_3P_3O_9$ | [55,60–63] |
| Pyrophosphate | PPi(V) | $K_4P_2O_7$ | [64] |
| Pyrophosphite | PPi (III) | $Na_2H_2P_2O_5$ | Has been detected in meteorites and can be generated from phosphite under hot acidic hydrothermal conditions; phosphate can be reduced to phosphite by serpentinization [65–69] |
| Phosphoenolpyruvate | PEP | $KC_3H_5O_6P$ | Has the highest phosphoryl-transfer potential found in living organisms ($\Delta G^{o'} = -62$ kJ/mol) [70], and is an intermediate in gluconeogenesis and glycolysis, where its conversion to pyruvic acid by pyruvate kinase generates ATP via substrate-level phosphorylation |
| Carbamoyl phosphate | CP | $Li_2CH_2NO_5P \cdot xH_2O$ | Can be made abiotically and has a role in extant biochemistry [71] |
| Trimethyl phosphate | TMP | $(CH_3)_3PO_4$ | Has been studied for its potential role in the nonenzymatic conversion of hypoxanthine to adenine [72] |

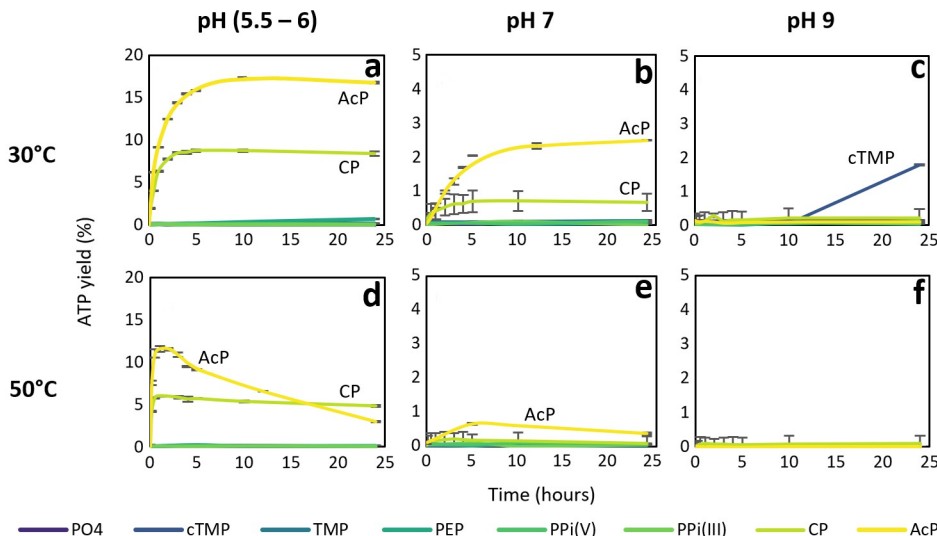

**Fig 3. ATP synthesis with different phosphorylating agents.** 1 mM:4 mM ADP:phosphorylating agent reaction catalysed by $Fe^{3+}$ with various phosphorylating agents at different pH and temperature. AcP, acetyl phosphate; CP, carbamoyl phosphate; cTMP, trimetaphosphate; PEP, phosphoenolpyruvate; $PO_4$: potassium phosphate; PPi(III), pyrophosphite; PPi(V), pyrophosphate; TMP, trimethyl phosphate. $N = 3$ ±SD. (**a**) ATP yield (%) over 24 h at pH approximately 5.5–6 and 30°C; (**b**) ATP yield (%) over 24 h at pH 7 and 30°C; (**c**) ATP yield (%) over 24 h at pH 9 and 30°C; (**d**) ATP yield (%) over 24 h at pH approximately 5.5–6 and 50°C; (**e**) ATP yield (%) over 24 h at pH 7 and 50°C; (**f**) ATP yield (%) over 24 h at pH 9 and 50°C. The data underlying this figure can be found in Table H–M in S1 Data (sheet 3).

## Phosphorylation of ADP to ATP is unique among nucleoside diphosphates

We next explored the propensity of AcP to phosphorylate other canonical nucleoside diphosphates (NDPs), specifically cytidine diphosphate (CDP), guanosine diphosphate (GDP), uridine diphosphate (UDP), and inosine diphosphate (IDP). While not a canonical base, inosine is the precursor to both adenosine and guanosine in purine synthesis. Importantly, from a mechanistic point of view, inosine lacks the amino group incorporated at different positions onto the purine rings of adenosine and guanosine, but like GDP, IDP has an oxygen in place of the N6 amino group of adenosine. The results clearly show that AcP will phosphorylate ADP but not other NDPs (**Fig 4A–4E**), demonstrating a strong dependence on the structure of the nucleobase. For all NDPs, a peak for the corresponding triphosphate was present at the start of the reaction, but this did not change over 3 h for any NDP except ADP. As noted above for ADP, the presence of the NTP at 0 h can be ascribed to minor contamination of the commercial standard during the manufacturing process (more striking in the case of ATP). It was also striking that although AcP could also phosphorylate AMP to ADP (**Fig 4F**) the yields were much lower than the phosphorylation of ADP to ATP, suggesting that $Fe^{3+}$ interacts most strongly with the beta phosphate of ADP, and much less strongly with the alpha phosphate.

The fact that neither pyrimidine NDP could be phosphorylated suggests that the purine ring (or at least adenosine) is essential for positioning the interactions between $Fe^{3+}$ and AcP. ADP has an amine group at N6, whereas GDP has a carbonyl at C6 and an amine group at N2; inosine has a carbonyl group at C6; and both GDP and IDP have a protonated N at N1. It is possible that the N6 amino group of adenosine interacts with the carboxylate oxygen on AcP (see below), but otherwise, it is difficult to identify a specific mechanism from these results, as the N7 nitrogen, which is known to interact with metal ions, is equivalent in all 3 purine rings

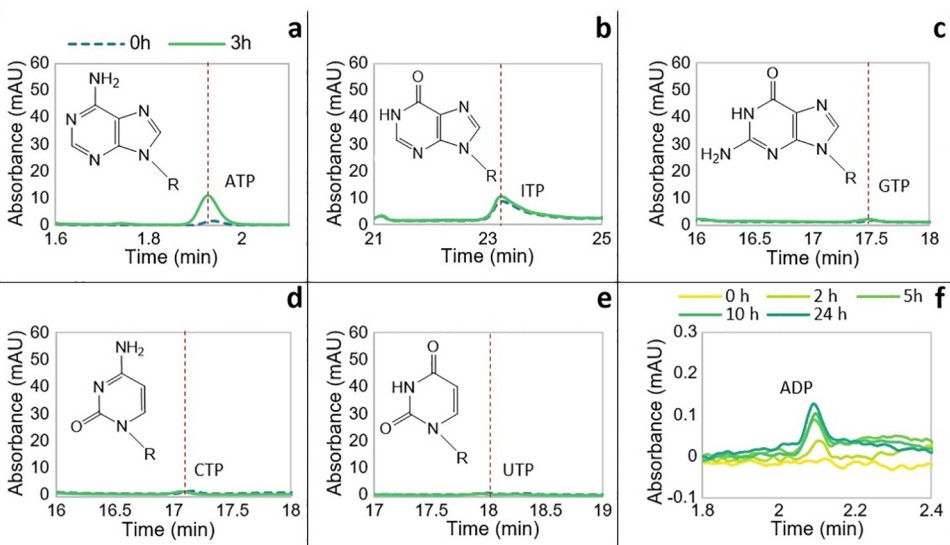

**Fig 4. Phosphorylation of NDPs by AcP.** HPLC chromatogram of the resulting NTP of the phosphorylation of (**a**) ADP, (**b**) IDP, (**c**) GDP, (**d**) CDP, and (**e**) UDP by AcP catalysed by $Fe^{3+}$ at 30°C and pH approximately 5.5–6 at the beginning of the reaction (0 h, broken line, teal) and after 3 h (solid line, green). The molecular structure of each nucleobase forming the nucleotides is shown. (**f**) HPLC chromatogram showing the progressive ADP synthesis over 24 h via phosphorylation of AMP by AcP in the presence of $Fe^{3+}$ at 30°C. The data underlying this figure can be found in Table N–S in S1 Data (sheet 4). AcP, acetyl phosphate; CDP, cytidine diphosphate; GDP, guanosine diphosphate; IDP, inosine diphosphate; NDP, nucleoside diphosphate; UDP, uridine diphosphate.

[76–79]. Nonetheless, the N6 amine group clearly influences electron delocalisation in purine rings and, therefore, the basicity of the other nitrogen moieties. We return to this in MD simulations below.

## Catalysis of ADP phosphorylation does not involve nucleotide stacking

To understand how $Fe^{3+}$ catalyses the phosphorylation of ADP to ATP, we tested the effect of varying the $Fe^{3+}$ ion concentration. Holding the ADP and AcP concentrations constant at 1 mM and 4 mM, respectively, we varied the $Fe^{3+}$ concentration from 0.05 to 2 mM. We found that the maximal ATP yield was produced by 1 mM $Fe^{3+}$, indicating that the optimal $ADP:Fe^{3+}$ stoichiometry of the reaction was 1:1 (**Fig 5A**). Following Kitani and colleagues [52], we confirmed that low concentrations of either $Mg^{2+}$ or $Ca^{2+}$ (up to 2 mM) slightly increased the ATP yield in the presence of 1 mM $Fe^{3+}$. This suggests that either of these divalent cations can stabilise the newly formed ATP against hydrolysis and possibly liberate $Fe^{3+}$ to catalyse the next phosphorylation of ADP (**Fig 5A**). We note that chelation of $Fe^{3+}$ by ligands such as EDTA prevented the phosphorylation reaction altogether (**Fig E in S1 Results**), as did the use of clusters such as 4Fe4S clusters (**Fig B in S1 Results**) or hematite (**Fig 1**).

We next conducted a kinetic study of the phosphorylation reaction, specifically varying the ADP concentration and monitoring the reaction rate over the first 5 h of reaction (i.e., until the ATP yield starts to plateau, as in **Fig 2**), using the first-order rate equation $r = \Delta[ATP]/\Delta t$. The resulting curve resembled a characteristic Michaelis–Menten mechanism for an enzyme, indicating that $Fe^{3+}$ does indeed act as a catalyst (**Fig 5B**). The question remained whether a single $Fe^{3+}$ was interacting directly with a single ADP and AcP, or whether larger units such as stacked ADP rings were involved. Stacking can alter the geometry of which group interacts with $Fe^{3+}$ (**Fig F in S1 Results**) and has previously been suggested as a possible mechanism [80]. However, MALDI-ToF analysis, which can sensitively detect stacked nucleotides, showed

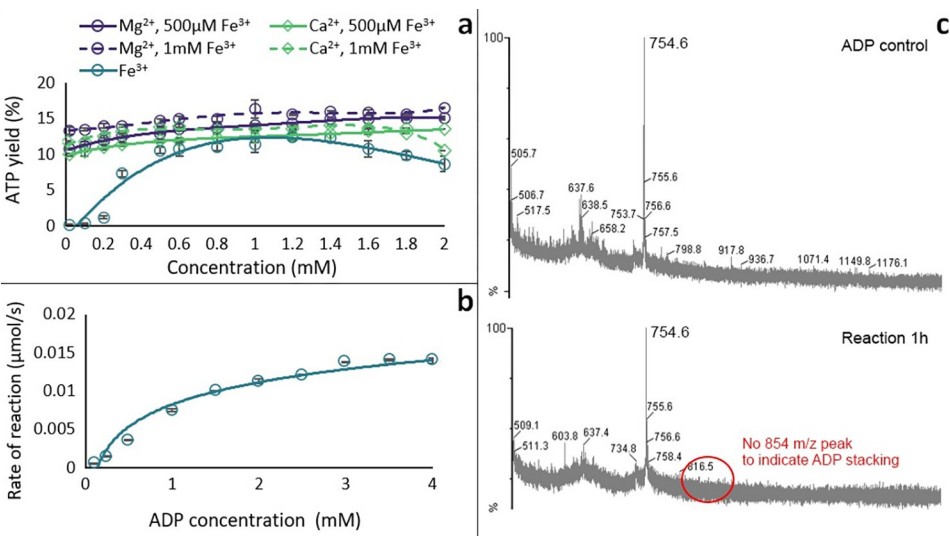

**Fig 5. Mechanism studies.** (**a**) Effect of varying concentration of $Fe^{3+}$ (teal circles) and adding increasing concentrations of $Mg^{2+}$ (purple circles) and $Ca^{2+}$ (green diamonds) on ATP yield at 2 h from the reaction ADP (1 mM) + AcP (4 mM) with 0.5 mM $Fe^{3+}$ (solid line) and 1 mM $Fe^{3+}$ (broken line) at 30°C and pH approximately 5.5–6. ($N = 3$ ±SD and 2 ±SD, respectively). (**b**) Michaelis–Menten kinetic analysis on the ADP + AcP reaction catalysed by $Fe^{3+}$ (0.5 mM). $N = 3$ ±SD. (**c**) MALDI-ToF spectra of ADP control (top) and a reaction sample at 1 h (bottom). The data underlying this figure can be found in Table T and U and Fig A in S1 Data (sheet 5).

no difference between the ADP control and the reaction sample; the main visible peaks appeared to be dimers of ADP/AMP present in the commercial ADP standard, possibly due to freeze-drying during production of ADP [81] (**Fig 5C**). While it is possible that this negative result reflects an issue with the MALDI-Tof analysis, we have previously detected stacks of AMP monomers using the same instrument and methodology [15]. It therefore seems likely that the difference in this case reflects the use of ADP rather than AMP.

The tentative conclusion that stacking of bases does not facilitate ADP phosphorylation was supported by MD simulations (**Fig 6**). These do not simulate chemical reactions but rather the forces acting between atoms and molecules under equivalent simulated experimental conditions. MD simulations can therefore predict the proportion of the simulation time in which specific atoms are oriented within plausible reaction distance [82]. For AcP to phosphorylate ADP to ATP, the phosphate of AcP needs to be positioned within a few Ångstroms of the beta phosphate oxygens of ADP. This is clearly seen in **Fig 6A**, where there is a strong peak representing a favourable conformation within 5 Å (and a secondary peak around 5 Å) in which the bond-forming groups are oriented within a feasible reaction space. This orientation was seen in the presence $Fe^{3+}$ but was strongly subdued with $Fe^{2+}$ or $Mg^{2+}$ ions, indicating that only $Fe^{3+}$ had strong enough interactions with phosphate to bring the 2 phosphate groups into close proximity for extended periods, corroborating our experimental findings. In this context, **Fig 6B** shows that a single ADP is clearly better than 2 (potentially stacked) ADPs in the same system. While the molar ratios have been kept consistent, the scope for interactions between 2 ADPs clearly suppressed the interaction time between the beta phosphate of any ADP and the AcP phosphate. This corroborates our MALDI-ToF results, as both sets of data indicate that the reaction between ADP and AcP does not occur between stacked dimers but rather between monomers. The bottom panels are consistent with these findings, showing that only $Fe^{3+}$ ions spend a significant proportion of the simulation time positioned close to the ADP beta phosphate (**Fig 6C**) or the AcP phosphate (**Fig 6D**), thereby positioning them suitably close to react.

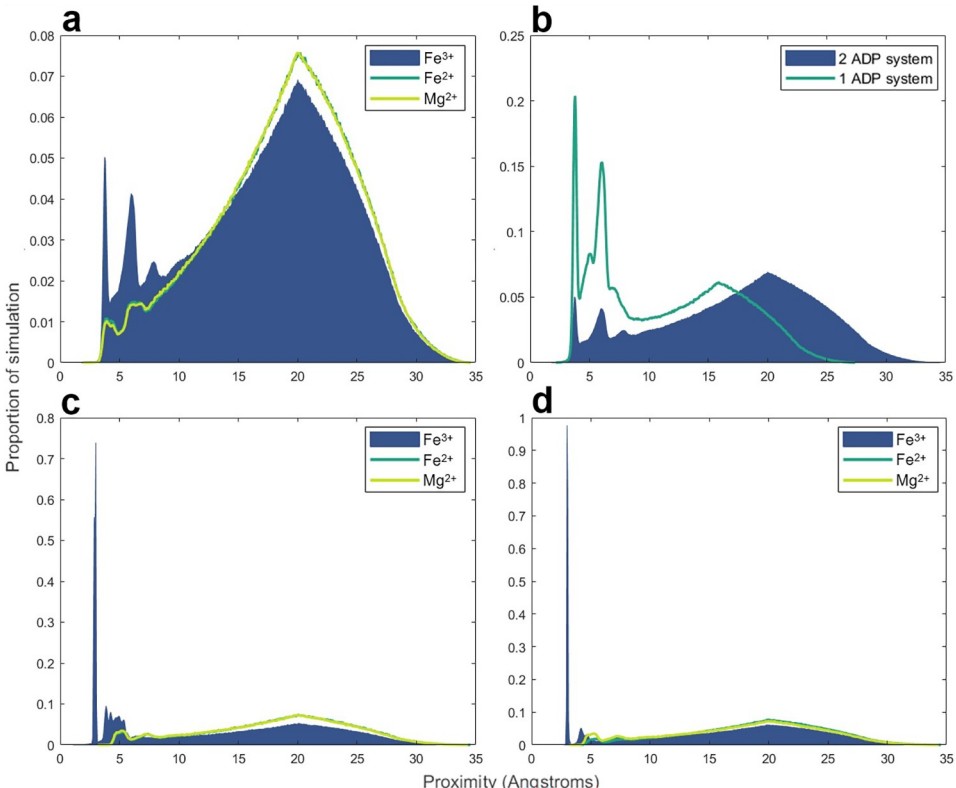

**Fig 6. MD simulations. (a)**. Proximity of AcP phosphate to ADP beta phosphate oxygens in the presence of either $Fe^{3+}$, $Fe^{2+}$, or $Mg^{2+}$. System components: 2 ADP, 8 AcP, 8 $Li^+$, 4 $Na^+$, 12 $K^+$, 18 $Cl^-$, and either 2 $Fe^{3+}$, 2 $Fe^{2+}$, or 2 $Mg^{2+}$. ADP and AcP both have charges of 2−. (**b**). Proximity of AcP phosphate to ADP beta phosphate oxygens with either 1 or 2 ADPs with constant molecular and ionic ratios. The 2-ADP system is the same experiment as the ferric iron simulation from panel (a). The 1-ADP system uses the same molecule and ion ratios but with the total quantities halved. To maintain comparable concentrations, the periodic box size was reduced to 31.75 Å in the 1-ADP system. (**c**). Proximity of ADP beta phosphate to either $Fe^{3+}$, $Fe^{2+}$, or $Mg^{2+}$ in the simulations from panel (a). (**d**). Proximity of AcP phosphate to either $Fe^{3+}$, $Fe^{2+}$, or $Mg^{2+}$ in the simulations from panel (a). The data underlying this figure can be found in Table V–Y in S1 Data (sheet 6). AcP, acetyl phosphate; MD, molecular dynamic.

Our MD simulations also show striking differences between ADP and GDP in terms of the proportion of simulation time that $Fe^{3+}$ interacts with N atoms in the purine ring (**Fig G in S1 Results**). With the exception of the N7 nitrogen in ADP (**Fig Gc in S1 Results**), which is known from experimental work to interact strongly with $Fe^{3+}$ [76–79], facilitating phosphorylation, the other N atoms in the guanine ring interact more strongly with $Fe^{3+}$ than those in the adenine ring (**Fig Ga**, **Gb**, and **Gd in S1 Results**). This presumably has the effect of partially abstracting the catalyst, such that it is less available to interact with AcP or the beta phosphate on GDP than on ADP. Thus, MD simulations corroborate our experimental findings and show that $Fe^{3+}$ ions can interact with the N7 on the ADP ring as well as the AcP phosphate and the ADP beta phosphate, forming a macro-chelate complex that facilitates ATP synthesis.

These findings suggest that the high charge density of $Fe^{3+}$ allows it to interact directly with the N7 on the adenine ring, while anchoring AcP in position for its phosphate group to interact with the diphosphate tail of ADP, giving a taut conformation of ADP (**Fig 7A**). The interaction with the dianion has been proposed before [83,84] and is key because at the optimal pH of 5.5 to 6, the first 2 hydroxyl groups of ADP (p$K_a$ 0.9 and 2.8) are deprotonated, while the external OH group (p$K_a$ 6.8) remains protonated, and is therefore not available for nucleophilic

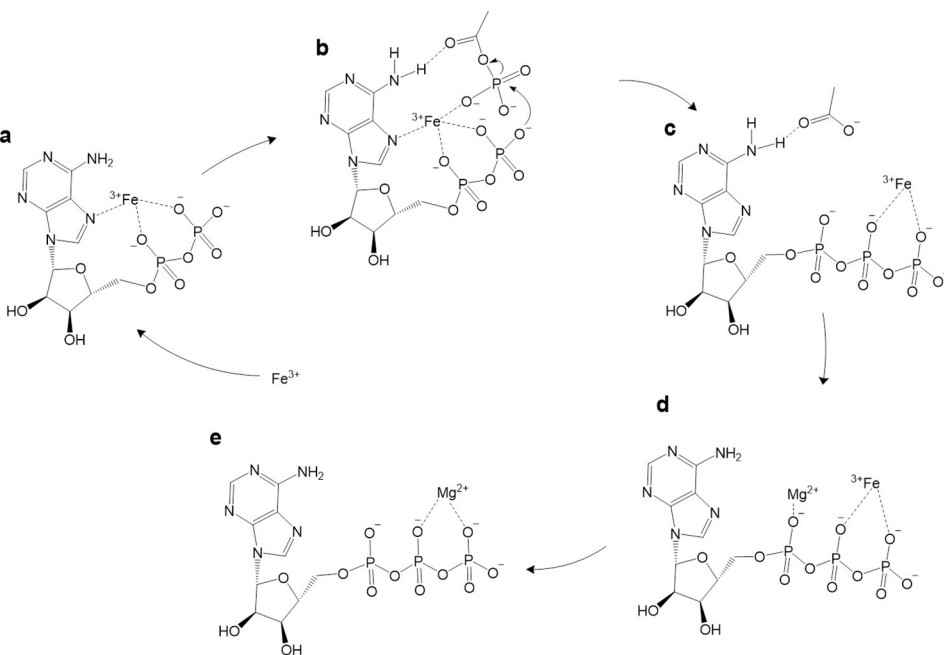

**Fig 7. Potential mechanism of ADP phosphorylation in water.** $Fe^{3+}$, stabilised by the N7 group on adenine, interacts with the dianion of ADP, lowering the $pK_a$ of the outermost OH group and enhancing its nucleophilicity (**a**). $Fe^{3+}$ interacts with the oxygens of a molecule of the surrounding AcP, bringing it close enough to facilitate the phosphate transfer (**b**). This interaction might be stabilised by further interactions between the N6 amino group and the carboxylate oxygen on AcP (**b**). Note that we only depict interactions between the $Fe^{3+}$ and moieties on the ADP, adenosine ring, and AcP that have been established by our experiments and MD simulations, and this is not intended as a full depiction of the coordination sphere; for clarity, we do not include any interactions with water, which would certainly be part of the coordination sphere. $Fe^{3+}$ is then likely to move from Pα to the Pβ and Pγ of ATP, although this is uncertain (**c**), and ultimately abandons the ATP chelated by acetate groups facilitated by the favourable association of $Mg^{2+}$ as suggested by the reaction kinetics (**d**). $Fe^{3+}$ is then available to catalyse another phosphorylation of ADP (**e**). AcP, acetyl phosphate; MD, molecular dynamic.

attack [85]. The interaction of the 2 deprotonated OH groups with $Fe^{3+}$ has the effect of lowering the $pK_a$ of the outermost OH group, thus deprotonating it and enhancing its nucleophilicity (**Fig 6B**). Possibly stabilised by interactions between the carboxylate oxygen of AcP and the N6 amino group on the adenosine ring (**Fig 7B**), the phosphate group of AcP is now readily positioned for nucleophilic attack by the newly deprotonated $O^-$ of ADP, forming ATP (**Fig 7C**). This mechanism might also help explain why $Ca^{2+}$ and $Mg^{2+}$ slightly increase the rate of reaction; these ions could displace $Fe^{3+}$ from the ATP product (as they interact better with the triphosphate tail; **Fig 7D**) or alternatively protect newly formed ATP from hydrolysis after displacement of $Fe^{3+}$. Given the approximation to Michaelis–Menten reaction kinetics, we assume that $Fe^{3+}$ is displaced from the ATP, being freed to catalyse further rounds of ADP phosphorylation (**Fig 7E**).

## Discussion

Our results support the following conclusions: (i) AcP efficiently phosphorylates ADP to ATP, but only in the presence of $Fe^{3+}$ ions as catalyst (**Fig 1**); (ii) the reaction takes place in water and can occur in a wide range of aqueous environments (**Fig 2**); (iii) no other phosphorylating agent tested was as effective as AcP (**Fig 3**); and (iv) adenine is unique among canonical nucleobases in facilitating the phosphorylation of its nucleoside diphosphate to the triphosphate (**Fig 4**). Taken together, these findings suggest that the preeminence of ATP in biology

might have its roots in aqueous prebiotic chemistry. The substrate-level phosphorylation of ADP to ATP by AcP is uniquely facilitated in water under prebiotic conditions and remains the fulcrum between thioester and phosphate metabolism in bacteria and archaea today [2]. This implies that ATP could have become the universal energy currency of life not as the end-point of genetic selection or as a frozen accident, but for fundamental chemical reasons, and arguably in a monomer world before the polymerization of RNA, DNA, and proteins.

The work presented here provides a compelling basis for each of these statements, but also raises a number of questions. Why ferric iron? Unlike AcP or ATP itself, there is no clear link with life in this case; we had expected other ions more commonly associated with nucleotides, notably $Mg^{2+}$ or $Ca^{2+}$ [54,55], to play a more clear-cut role. In fact, their catalytic effect was only noticeable in the presence of $Fe^{3+}$, as previously reported [52], whereas higher concentrations, equivalent to modern ocean conditions, precluded ATP synthesis. We infer that the reason $Fe^{3+}$ plays a unique role relates in part to its high charge density and small ionic radius. The fact that ADP is phosphorylated more readily than AMP (**Fig 4**) indicates that $Fe^{3+}$ interacts with the diphosphate tail of ADP, which is also borne out by molecular dynamic simulations (**Fig 6**). The fact that the optimal stoichiometry of $Fe^{3+}$ to ADP is 1:1, coupled with the absence of evidence for stacking of bases by MALDI-ToF (**Fig 5**), suggests that a single $Fe^{3+}$ ion interacts with a single ADP, and necessarily also with a single AcP. Again, this interpretation is supported by MD simulations, which show that interactions between 2 ADP molecules suppress their interactions with AcP (**Fig 6B**). Interestingly, the MD simulations also showed that with the exception of N7, $Fe^{3+}$ interacts more strongly with all the N atoms on the guanosine ring compared with those on the adenosine ring (**Fig G in S1 Results**). This finding is consistent with earlier work showing that the replacement of the N6 amine on the adenosine ring with the carbonyl group on the inosine and guanosine rings affects the properties of the other Ns on the purine ring, in particular their basicity [86]. Our results suggest that the stronger interactions between $Fe^{3+}$ and the guanosine Ns likely abstract the catalyst, hindering phosphorylation reactions.

As shown in **Fig 7**, these stipulations require a taut molecular configuration of ADP, which is far from the loose conformation usually depicted, if only for ease of presentation. The orientation of the adenine ring in nucleotides has long been disputed, with some arguing that it should face the opposite way, with a more "rigid" (*anti-gg*) conformation [87]. Others have suggested an equivalent orientation to that proposed here [84,88], some specifically with $Fe^{3+}$ [76,77]. Furthermore, a similar complex between ATP and $Cu^{2+}$ has been reported to efficiently catalyse Diels–Alder reactions, thus confirming that such an orientation of the molecule is possible [89]. In any case, this taut conformation almost certainly requires the interacting ion to have a high charge density and small ionic radius, to draw each of these groups into close enough proximity to react. Among the cations tested here, $Fe^{3+}$ has the highest charge density and the smallest ionic radius [90]. Nonetheless, some of the other ions studied, notably $Cr^{3+}$ and $Co^{3+}$, have a similar ionic radius and charge density, yet do not have a remotely comparable catalytic effect, so the size and charge density cannot be the only explanation for our results. The electronic configuration of $Fe^{3+}$ may also play a role: Unlike $Cr^{3+}$ and $Co^{3+}$, $Fe^{3+}$ has the electronic configuration $[Ar]3d^5$, having all 5d orbitals half occupied. However, $Mn^{2+}$, which can substitute $Mg^{2+}$ in the catalytic centre of acetate kinase, has an equivalent 3d orbital, yet yielded negative results in our experiments. If so, then size, charge density, and electronic configuration might all play a role, potentially by stabilizing the phosphorylation transition state as a macro-chelate complex with ADP and AcP [86,91,92]. This hypothesis needs to be explored in future work. A related question, given our emphasis on life as a guide to protometabolism, is why $Fe^{3+}$ is no longer used to catalyse ADP phosphorylation in biology. The most likely possibility is simply that $Fe^{3+}$ always had limited availability (but see below)

and undesirable reactivity, such as catalysing Fenton chemistry [93]. We note that chelated forms of $Fe^{3+}$ do not catalyse ADP phosphorylation—neither FeS clusters (**Fig B in S1 Results**) nor EDTA-chelated $Fe^{3+}$ (**Fig E in S1 Results**) generated any ATP at all. Nor did $Fe^{3+}$ on surfaces such as hematite (**Fig 1**). For these reasons, enzymatic catalysis most likely displaced $Fe^{3+}$ catalysis from later metabolism, especially given the high demand for ATP.

Why AcP? The idea that this small (2-carbon) molecule might have acted as a prebiotic phosphoryl donor has a long history, going back to Lipmann himself [15,36,38,42–48], as indeed does its confounding potential as an acetyl donor [15,37,94–97]. AcP still plays a global signalling and energy transduction role in bacteria [98], in part because its free energy of hydrolysis (and therefore its phosphorylating potential) is greater than that of ATP ($\Delta G^{o\prime}$ = −43 kJ $mol^{-1}$ versus −31 kJ $mol^{-1}$, respectively). When complexed in a 1:1 ratio with ADP, therefore, AcP has the potential to transfer its phosphate to form ATP, and so serves as a labile energy source in cells, linked to the excretion of acetate as waste. But the actual change in $\Delta G$ depends on how far from equilibrium the ratio of AcP/Ac + Pi or ATP/ADP + Pi has been pushed, and so varies depending on conditions. In our experiments, all phosphoryl donors were added at equivalent excess. The fact that the $\Delta G^{o\prime}$ for hydrolysis of PEP (−62 kJ $mol^{-1}$) and CP (−51 kJ $mol^{-1}$) are markedly greater than that for AcP means that free-energy change is only part of the explanation for the efficacy of AcP. As ATP was primarily formed by AcP in the presence of $Fe^{3+}$ ions, the critical factors may instead have been (i) the position of the 2 phosphoester oxygen atoms in relation to the $Fe^{3+}$; (ii) the phosphate group in relation to the diphosphate tail of ADP; and (iii) the carboxyl oxygen of AcP in relation to the N6 amine group of adenine (**Fig 7**). This latter point might also discriminate ADP from GDP. In other words, both AcP and ADP are favoured not for selective or thermodynamic reasons, but kinetic—because their chemistry is facilitated by molecular geometry in aqueous prebiotic environments.

The only other molecule with equivalent geometry in this regard is CP, which our model would therefore predict should have some phosphorylating efficacy. CP was indeed the only other species to show significant phosphorylating activity in our system (**Fig 3**). CP has long been considered a plausible prebiotic phosphorylating agent [71,99–101], albeit possibly in the form of cyanate and Pi, which is in equilibrium with CP [102–105]. CP can also promote the formation of ATP in the presence of $Ca^{2+}$ or $Ba^{2+}$ ions [71,106–108]. Like AcP, CP retains a place in modern metabolism, for example, as a substrate for carbamate kinase, phosphorylating ADP to ATP in microbial fermentation of arginine, agmatine, and oxalurate/allantoin [109], as well as the de novo synthesis of pyrimidines (although not as a phosphorylator) [110]. Taken together with our own results, these findings suggest that both AcP and CP are molecular "living fossils" of prebiotic chemistry, retaining a role in modern metabolism due to their felicitous aqueous chemistry (**Fig 3**). Nonetheless, in general, CP does not perform phosphorylation reactions in extant life, preferring primary N-carbamoylation reactions [111]. The only other phosphorylating agent that showed any phosphorylating potential in our panel was cTMP, which is known to self-condense from diamidophosphate (DAP) in solution [112]. DAP has previously been shown to phosphorylate NDPs to the triphosphates [112]. However, we have not pursued this further as phosphoamidates have little relevance to extant biology. No N–P bonds exist in central metabolism common to archaea and bacteria, and creatine phosphate is restricted to eukaryotes, so is unlikely to reflect a biomimetic protometabolism.

Surprisingly, our results demonstrate that maximal ATP synthesis occurred at high water activity and low ion concentrations, indicating that prebiotic ATP synthesis would be most feasible in freshwater systems. Likewise, ferrous iron can be oxidized to ferric iron by photochemical reactions or oxidants such as NO derived from volcanic emissions, meteorite impacts, or lightning strikes, which also points to terrestrial geothermal systems as a plausible environment for aqueous ATP synthesis [113,114]. Conversely, high concentrations of $Mg^{2+}$

(50 mM) and $Ca^{2+}$ (10 mM) precluded ATP synthesis, implying that this chemistry would not be favoured in modern oceans. Nonetheless, our results do not exclude submarine hydrothermal systems as potential environments for this chemistry. Some shallow submarine systems such as Strytan in Iceland are sustained by meteoritic water and feature $Na^+$ gradients as well as $H^+$ gradients [115]. Such mixed systems could have been common in shallow Hadean oceans. Moreover, the concentration of divalent cations in the Hadean oceans may have been lower than modern oceans, with estimates varying widely [55,116]. Regardless of mean ocean concentrations, alkaline hydrothermal systems tend to precipitate $Ca^{2+}$ and $Mg^{2+}$ ions as aragonite and brucite, so their concentrations are typically much lower than mean ocean values. Modelling work in relation to Hadean systems indicates that hydrothermal concentrations of $Ca^{2+}$ and $Mg^{2+}$ would likely have been <1 mM [117,118], which is in the range that enhanced phosphorylation here. Other conditions considered here, including salinity and high pressure [59], would have only limited effects on ATP synthesis in submarine hydrothermal systems (which typically have pressures in the range of 100 to 300 Bars). Alkaline hydrothermal systems might also have generated $Fe^{3+}$ in situ for ADP phosphorylation. Thermodynamic modelling shows that the mixing of alkaline hydrothermal fluids with seawater in submarine systems can promote continuous cycling between ferrous and ferric iron, potentially forming soluble hydrous ferric chlorides [118], which our experiments show have the same effect as ferric sulphate (**Fig H in S1 Results**). The availability of ferric iron is critical for other prebiotic catalysts including cysteine-FeS clusters [25,53,119,120] and has been discussed in more detail elsewhere [53].

A major question for prebiotic chemistry is how could an energy currency power work? As noted in the Introduction, there is nothing special about the bonds in ATP; rather, the ATP synthase powers a disequilibrium in the ratio of ADP to ATP, which amounts to 10 orders of magnitude from equilibrium in the cytosol of modern cells. Molecular engines such as the ATP synthase use ratchet-like mechanical mechanisms to convert environmental redox disequilibria into a highly skewed ratio of ADP to ATP [121]. But how could a simple prebiotic system composed mostly of monomers sustain a disequilibrium in ATP to ADP ratio that powers work? One possibility is that dynamic environments could sustain critical disequilibria across short distances such as protocell membranes. For example, alkaline hydrothermal systems sustain steep pH gradients across thin inorganic barriers, as mildly acidic Hadean ocean waters (pH 5 to 6) continually mix with strongly alkaline hydrothermal fluids (pH 9 to 11) in microporous labyrinths that operate as electrochemical flow reactors [20,47,122,123]. Protocells with mixed amphiphile membranes could bind to mineral barriers and potentially use these proton gradients to drive work, including ATP synthesis [124–127] Mixed amphiphile membranes are highly permeable to protons, because fatty acid flip flop continuously transfers protons from the acid exterior to the alkaline interior of protocells [124,128,129]. A continuous influx of protons across protocell membranes could in principle promote ADP phosphorylation under locally acidic conditions in the immediate vicinity of the membrane within protocells. The ATP formed inside protocells would then be more vulnerable to hydrolysis linked to phosphorylation under more alkaline conditions in the bulk water of the cytosol. At face value, the ATP yield reported here at pH 5.5 to 6 after 10 h was 17.4% (corresponding to 156.5 μM) while the yield at pH 9 was 0.043%, corresponding to 0.4 μM, a difference of 400-fold. Thus, a geologically sustained difference in pH across membranes could drive a local disequilibrium in the ATP/ADP ratio of 2 to 3 orders of magnitude, enough to power work even in the absence of other possible factors such as temperature. Higher temperatures (50˚C) promote both the rapid synthesis and hydrolysis of ATP (**Fig 2B**), which should amplify this driving force. We stress that these considerations require further elucidation, but in principle, steep pH gradients can drive a disequilibrium in the ATP/ADP ratio that could power work.

What would constitute "work" under these conditions? We are thinking in particular of the coupling of exergonic ATP hydrolysis to endergonic reactions in protometabolism, most pertinently the polymerization of nucleotides, amino acids, or both. ATP itself is relatively stable and hence is unlikely to adenylate or phosphorylate other molecules in the absence of suitable catalysts [130]. Which catalysts would best promote ATP hydrolysis coupled to phosphorylation in the aqueous prebiotic environments discussed here is a separate question that will be addressed elsewhere; metal ions, amino acids, short peptides, short ribozymes, or nucleotide cofactors all deserve consideration. We suspect that ATP only displaced earlier phosphorylating agents such as AcP or CP when coupled by catalysts to polymerization reactions linked to the hydrolysis of pyrophosphate, as discussed in the Introduction, or potentially as an autocatalytic feedback loop in purine synthesis, which could have amplified purine nucleotide availability in early evolution [18,129]. Nonetheless, the work presented here shows that ATP might have entered protometabolism at an earlier stage than generally supposed and could have contributed to the transition from a monomer to a polymer world.

In conclusion, AcP is unique among a panel of relevant phosphorylating agents in that it can phosphorylate ADP to ATP, in water, in the presence of $Fe^{3+}$. AcP is formed readily through prebiotic chemistry and remains central to prokaryotic metabolism, making it the most plausible precursor to ATP as a biochemical phosphorylator [15]. Critically, AcP does not phosphorylate other nucleoside diphosphates, giving a compelling new insight into how ATP might have come to be so dominant in modern metabolism. Our findings indicate that the high charge density and electronic configuration of $Fe^{3+}$ can position molecules in water to react in the absence of macromolecular catalysts such as RNA or proteins, or even mineral surfaces. Beyond that, our results suggest that steep pH gradients could in principle generate disequilibria in the ratio of ATP to ADP of several orders of magnitude, enabling ATP to drive work even in a prebiotic monomer world. Given suitable catalysts, ATP could eventually displace earlier phosphorylating agents and promote the polymerization of amino acids and nucleotides to form RNA, DNA, and proteins by liberation of pyrophosphate as a leaving group. If so, then ATP became established as the universal energy currency for reasons of prebiotic chemistry before the emergence of genetically encoded macromolecular engines.

## Materials and methods

### Materials

All salts were purchased from Sigma-Aldrich, except for copper nitrate hemipentahydrate (Cu$(NO_3)_2 \cdot 2.5H_2O$), copper sulphate pentahydrate ($CuSO_4 \cdot 5H_2O$) and manganese nitrate hexahydrate ($Mn(NO_3)_2 \cdot 6H_2O$, Alfa Aesar), TEAA (triethylammonium acetate, Fluka), and CTP (Cytidine 5′-triphosphate sodium salt, Cambridge Bioscience). All solvents were HPLC-grade and purchased from Fischer. All reagents used were analytical grade ($\geq$96%).

**Reaction setup.** Depending on the solubility of the analytes, reactions were carried out in either a stationary (SciQuip HP120-S) or a shaking (ThermoMix HM100-Pro) dry block heater.

For the reaction, stock solutions of di-nucleotides (sodium salts, $\geq$96%, Sigma-Aldrich), phosphorylating agents, and metal catalyst were freshly prepared as to avoid freeze-thawing (10 mM for reactions to be analysed via HPLC, 1 M for reactions to be analysed via NMR). Except where indicated, the ratios of analytes in a solution were 1(ADP):4(AcP) and 1($Fe^{3+}$):2 (ADP). When needed, the pH was adjusted using aqueous HCL and NaOH (1 M or 3 M)

After checking the pH (Fisher Scientific accumet AE150 meter with VWR semi-micro pH electrode), samples were taken at time points (0, 10, and 30 min, 1 to 5, 10, and 24h) and, unless otherwise specified, immediately frozen at −80˚C for next-day analysis.

### Pressure reactor

Experiments under pressure were performed in a pressure vessel (Series 4600-1L-VGR with single inlet valve, Parr Instrument Company), pressurised with $N_2$ gas and placed on a hotplate (Fisherbrand Isotemp Digital Stirring Hotplate) at 30°C. Samples for both the high pressure experiment and ambient pressure control experiment were prepared in 2 mL glass headspace vials (Agilent Technologies) whose caps were pierced with a needle.

### FeS clusters

FeS clusters coordinated by 5 mM of L-cysteine were prepared under anaerobic conditions, and water sparged with $N_2$ was used to prepare all solutions. Stock solutions of 10 mM $Na_2S$, 10 mM $FeCl_3$, 50 mM of L-cysteine, and 1 M of NaOH were prepared either in water or in 10 mM bicarbonate buffer (pH 9.1). A volume of 4 mL of $Na_2S$ and 4 mL of L-cys were added to 28 mL of water/buffer, and the pH adjusted to approximately 9.8 using NaOH. A volume of 4 mL of $FeCl_3$ was then added and the volume adjusted to 40 mL to obtain a 1-mM FeS solution.

Oxygen levels in the anaerobic glovebox were maintained below 5 ppm when possible, and no work was conducted if this level was surpassed.

### UV/Vis spectroscopy

UV/Vis spectroscopy was used to verify the formation of FeS clusters. A volume of 1 mL of FeS stock solution was placed in a crystal cuvette, which was sealed with parafilm under anaerobic conditions. Spectra were obtained using a Thermofisher NanoDrop 2000c, with a baseline correction of 800 nm.

### Analysis

**HPLC.** Samples were prepared at collection by spinning at 4,000 rpm for 2 min and diluting 200 μL in 800 μL of EDTA solution (500 μL in 100 mM $PO_4$ buffer at pH 7.1) prior to freezing, in order to chelate the $Fe^{3+}$ ions in solution that would otherwise block the HPLC column.

Thawed samples were filtered using syringe filters (ANP1322, 0.22 μm PTFE Syringe filter, Gilson Scientific) attached to a 1-mL sterile syringe (BD Plastipak Syringes) in 2 mL headspace vials and analysed on an HPLC instrument (Agilent Technologies, 1260 Infinity II); peaks were identified using pure standards. The wavelengths for UV detection were usually set at 254 nm and 260 nm (most suitable for cyclic rings such as adenosine), while the column tray temperature was maintained at room temperature. Two different columns were used depending on the pH of the sample being analysed: Poroshell 120 EC-C18 for pH 2 to 8 and Poroshell HPH-C18 for pH 9 to 11.

Mobile phase A consisted of 80 mM phosphate buffer (made by mixing equal parts of potassium phosphate dibasic (40 mM) and potassium phosphate monobasic (40 mM) salts dissolved in water) adjusted to pH 5.8 using 3 M HCl and filtered with 0.2 μm nylon membrane filters (GNWP04700, 0.2 μm pore size, Merck Millipore), while mobile phase B consisted of 100% methanol. The injection volume was 1 μL, with a flow rate of 1 mL/min, and the run was an isocratic gradient that consisted of 95% B for 5 min.

For experiments using nucleoside diphosphates with different bases, analyses were carried out on a Polaris C18-A column, with mobile phase A consisting of 10 mM potassium phosphate monobasic buffer with 10 mM tetrabutylammonium hydroxide (TBAH) adjusted to pH 8 using 3 M HCl and filtered with 0.2 μm nylon membrane filters (GNWP04700, 0.2 μm pore size, Merck Millipore), while mobile phase B consisted of 100% methanol (method described

**Table 2. HPLC method for G, C, I, and U nucleotides experiments.**

| | |
|---|---|
| Mobile phase A | 10 mM $KH_2PO_4$ + 10 mM TBAH in HPLC-grade water |
| Mobile phase B | 100% HPLC-grade methanol |
| Gradient | 5% B → 50% B (up during 25 min) → 50% B (for 2 min) → 95% B (up during 6 s) → 95% B (for 3 min) → 5% B (down during 6 s) → 5% B (for 2 min) |
| Flow rate | 1.5 mL/min |
| Injection volume | 1 μL |

in Table 2). The wavelengths for UV detection were set at 254, 260, and 271 nm for guanosine, uridine and inosine, and cytidine, respectively.

Two flush methods (Table 3) were employed to preserve the column: Flush 1 was used every 12 to 15 samples, then 3 rounds of Flush 1 followed by 1 run of Flush 2 were run prior to switching off the machine.

Computational analysis was done using Agilent OpenLAB software (ChemStation Edition). Each peak was manually integrated using the calibration curves as reference and the raw file was exported for data manipulation. As residual ATP is present in the ADP commercial standard, the yield of the reaction is calculated by subtracting the reading for ATP at time point 0 from all subsequent time point readings. Except where noted, the rate of the reaction was calculated over the first hour of reaction (to enable direct comparison between 30˚C and 50˚C) using the first-order rate equation r = Δ[ATP]/Δt.

**ESI MS.** Electrospray ionisation mass spectrometry was used to confirm the identity of ATP through MS/MS. After purification through SPE (see previous section), the sample was loaded into a 0.5-mL glass syringe (Gastight Syringe Model 1750 RN, Hamilton) and directly infused into the mass spectrometer (Finnigan LTQ Linear Ion Trap mass spectrometer) at a flow rate of 10 μL/min. To avoid contaminations, the syringe and line were flushed with 100% methanol before and after sample infusion, and the spectra recorded.

The mass spectrometer was operated in negative ion mode, and the capillary voltage was set at −16 V. Data were collected from 100 to 2,000 *m/z* with an acquisition rate of 5 spectra per second. For the MS/MS, Ar was used as the collision gas and the collision energy was adjusted to 30 eV. The software Xcalibur (Thermo Scientific) was used for method setup and data processing.

**MALDI-ToF MS.** Samples were thawed and desalted using a protocol adapted from Burcar and colleagues [131]. Two solvents were prepared: an ACN solution consisting of 50% acetonitrile in water and a 0.1-M TEAA solution in water.

Using a Millipore C18 zip tip (Sigma), 10 μL of ACN solution were aspirated and discarded 3 times. The 3 rinses were repeated with 10 μL of the TEAA solution. To allow for the retention

**Table 3. HPLC flush methods.** These are run at the end of a set number of sample analyses.

| | *FLUSH 1* | *FLUSH 2* |
|---|---|---|
| Mobile phase A | HPLC-grade water | HPLC-grade water |
| Mobile phase B | 100% HPLC-grade methanol | 100% HPLC-grade methanol |
| Gradient | 5% → 95% B (up during 15 min) → 95% B (for 5 min) → 5% B (down during 10 min) | Initial: 5% B (for 17 min) → 95% B (up during 18 min) → 95% B (for 17 min) → 60% B (down during 6 min) → 60% B (for 17 min) |
| Flow rate | 1 mL/min | 1 mL/min |

of the analyte by the zip tip matrix, 10 μL of sample were aspirated up and down 8 times and then discarded. A volume of 10 μL of water were aspirated and discarded, followed by 10 μL of the TEAA solution and once again 10 μL of water. A volume of 4 μL of ACN were slowly aspirated up and down 3 times and deposited into a small Eppendorf microcentrifuge tube.

The MALDI-ToF protocol used was designed by Whicher and colleagues [15]. The matrix consisted of 2,4,6-trihydroxyacetophenone monohydrate (THAP) and ammonium citrate dibasic and was freshly prepared before the analysis using equal volumes of stocks that were maintained at 4°C for a maximum of a week.

A volume of 2 μL of matrix solution was mixed with 2 μL of sample, deposited onto a clean steel MALDI-ToF plate and allowed to evaporate for 30 min before the introduction of the steel plate into the instrument (Waters micro MX mass spectrometer). The analytical conditions were as follows: reflectron and negative ion mode, 400 au of laser power, 2,000 V of pulse, 2,500 V of the detector, 12,000 V of flight tube, 5,200 V of reflector, 3,738 V of negative anode, and 500 to 5,000 amu of scan range. The mass spectrometer was calibrated using a low-molecular weight oligonucleotide standard (comprising of a DNA 4-mer, 5-mer, 7-mer, 9-mer, and 11-mer (Bruker Daltonics)). Each oligonucleotide standard was initially dissolved in 100 μL water, divided in aliquots and frozen at −80°C. A fresh aliquot was used at each analytical calibration.

## Molecular dynamics simulations

Simulation inputs were prepared using CHARMM-GUI's ligand modeller and multicomponent assembler [132,133]. The simulation parameters utilized a 40-Å periodic box, 25°C NVT ensemble, CHARM-GUI's automatic PME FFT grid information generation, Monte Carlo ion placement, and the CHARMM36m forcefield [134]. The system components were 2 ADP, 8 AcP, 8 $Li^+$, 4 $Na^+$, 12 $K^+$, 18 $Cl^-$, and either 2 $Fe^{3+}$, 2 $Fe^{2+}$, or 2 $Mg^{2+}$. ADP and AcP both have protonation states of −2, but as noted previously, the proximity of $Fe^{3+}$ lowers the pKa of the outermost OH group, resulting in its deprotonation and increasing its nucleophilicity. We did not explicitly model this in the MD simulations as this would require quantum simulations of the reactions themselves. For the 1-ADP experiment, the same molecule and ion ratios were used but with the total quantities halved. To maintain comparable concentrations in this simulation, the periodic box size was reduced to 31.75 Å. Simulations were run using NAMD 2 [135] on UCL's myriad cluster. Simulations were minimized and equilibrated for CHARMM-GUI's default number of timesteps, 10,000 and 125,000, respectively (500 timesteps = 1 ps), before being run for 48 h using 2 MPI cores, 200 times in parallel. Each parallel repeat had randomized initial velocities with constant starting positions. This produced simulations on approximately 1.5 microsecond timescales. The data from the simulation trajectories were extracted using the MDToolbox package for MATLAB [136] and analysed in MATLAB R2020b.

## Supporting information

**S1 Results. Document reporting all Supporting information figures.**
(DOCX)

**S1 Data. Excel document detailing raw data for all analyses.**
(XLSX)

## Author Contributions

**Conceptualization:** Silvana Pinna, Sean F. Jordan, John Ward, Finn Werner, Nick Lane.

**Data curation:** Silvana Pinna.

**Formal analysis:** Silvana Pinna, Aaron Halpern, Nick Lane.

**Funding acquisition:** John Ward, Finn Werner, Nick Lane.

**Investigation:** Silvana Pinna, Cäcilia Kunz, Aaron Halpern, Stuart A. Harrison.

**Methodology:** Silvana Pinna, Aaron Halpern, Stuart A. Harrison, Sean F. Jordan, John Ward, Finn Werner, Nick Lane.

**Supervision:** Sean F. Jordan, John Ward, Finn Werner, Nick Lane.

**Visualization:** Silvana Pinna, Aaron Halpern, Stuart A. Harrison.

**Writing – original draft:** Silvana Pinna, Nick Lane.

**Writing – review & editing:** Silvana Pinna, Cäcilia Kunz, Aaron Halpern, Stuart A. Harrison, Sean F. Jordan, John Ward, Finn Werner, Nick Lane.

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
