## [Editor Report · Decision Letter 0]

30 Sep 2021

Dear Nick, 

Thank you for submitting your manuscript entitled "A prebiotic basis for ATP as the universal energy currency" for consideration as a Research Article by PLOS Biology.

Your manuscript has now been evaluated by the PLOS Biology editorial staff, as well as by an academic editor with relevant expertise, and I'm writing to let you know that we would like to send your submission out for external peer review.

Please re-submit your manuscript within two working days, i.e. by Oct 04 2021 11:59PM.

Cheers,

Roli

Roland Roberts

Senior Editor

PLOS Biology

rroberts@plos.org

---

## [Decision Letter · Decision Letter 1]

13 Dec 2021

Dear Nick,

Thank you for submitting your manuscript "A prebiotic basis for ATP as the universal energy currency" for consideration as a Research Article at PLOS Biology. Your manuscript has been evaluated by the PLOS Biology editors, an Academic Editor with relevant expertise, and by four independent reviewers. Thanks for your patience while we discussed the decision with the Academic Editor.

You'll see that while the reviewers are intrigued by your findings, they raise a series of significant concerns (some overlapping), which will need to be addressed before further consideration. The Academic Editor also kindly provided some additional advice, and I've included an edited version of his/her comments at the foot of this letter; these don't contain any further requests, but they do contextualise the requests from the reviewers, emphasising some, and explicitly excusing you one request (the EPR).

In light of the reviews (below), we will not be able to accept the current version of the manuscript, but we would welcome re-submission of a much-revised version that takes into account the reviewers' comments. We cannot make any decision about publication until we have seen the revised manuscript and your response to the reviewers' comments. Your revised manuscript is also likely to be sent for further evaluation by the reviewers.

We expect to receive your revised manuscript within 3 months. 

**IMPORTANT - SUBMITTING YOUR REVISION**

*Re-submission Checklist*

*Published Peer Review*

*PLOS Data Policy*

*Blot and Gel Data Policy*

Sincerely,

Roli

Roland Roberts

Senior Editor

PLOS Biology

rroberts@plos.org

REVIEWERS' COMMENTS:

Reviewer #1:

[see attachment for fully formatted version]

The manuscript by Pinna et al. reports on the detailed characterization of the phosphorylation of ADP by acetyl phosphate (AcP) to form ATP in the presence of ferric ions. This previously observed reaction appears to be surprisingly specific as no other NDPs or transition metal ions can carry out equivalent reactions to any significant extent. ADP phosphorylation appears to be optimal at 30°C, mildly acidic pH and low salt. The authors provide explanations for their observations within the context of their already well-established hypothesis of life's origins in a hydrothermal vent. Besides the fact that, taken face value, the observations reported here would rather favor a surface pond or lake as the cradle of life, some of their propositions need to be further substantiated.

Page 3 "ATP in a prebiotic, monomeric world". This assumes that life originated in bulk water. But, if, instead, life first appeared on mineral surfaces then the needed energy could have been provided through the oxidation of H2 and CO by, for instance, catalytic FeNiS minerals. Only when more evolved protocells liberated themselves from those surfaces, and moved to bulk water, would have ATP become essential.

 "Even if only one nucleotide triphosphate can be dominant, the implication of a frozen accident is not a satisfying explanation." Why not? There are different levels of "frozen accidents". This has been discussed by Brandon Carter who, in 1984, proposed that the evolution towards intelligent life required 6 "hard steps", the first one being abiogenesis. Each step was statistically postulated to last between 600 and 800 million years. Although these periods of time may be considered workable in astronomy and geology, in biology they do not make the same sense. In the latter case it is necessary to invoke the occurrence of extremely unlikely (highly contingent) events to explain that time duration. They can be indeed defined as "frozen accidents".

Page 13. "Such steep pH gradients could in principle operate across protocells as well as inorganic barriers." The authors propose that ATP would have been synthesized outside protocells, under acidic conditions, to be then transported across a proto-membrane into a cell where, under alkaline conditions, it would have phosphorylated some substrate(s). Have they considered the implications of such a mechanism in terms of ATP/ADP concentrations and diffusion? ATP would have to diffuse towards the protocell and ADP do it in the opposite direction, without risking to disappear into the bulk water. Another crucial point is membrane permeability. As F. H. Westheimer wrote in 1987: "Phosphoric acid is specially adapted for its role in nucleic acids because it can link two nucleotides and still ionize; the resulting negative charge serves both to stabilize the diesters against hydrolysis and to retain the molecules within a lipid membrane. A similar explanation for stability and retention also holds for phosphates that are intermediary metabolites and for phosphates that serve as energy sources." So, one very important role of even proto-membranes should have been to keep charged metabolites inside the cell to sustain metabolism. This, in turn, would have made ADP diffusion out of the protocell difficult to envision. 

Page 15. "…enabling ATP to drive work even in a prebiotic monomer world." How are the authors defining 'work' in that setting? In contemporary biology work is i) performed mechanically as in muscle contraction and ii) against a gradient of chemical potential in active transport (see Jencks, 1989). This work depends on complex protein-based binding energy modulation and the coupling of intermediate reaction steps. Of course, this kind of work could not have existed prebiotically. At that time the most likely role of ATP (if it already existed) would have been , for instance, phosphorylation to render carbonyl carbons electrophilic enough to be attacked by a nucleophile such as ammonia. It is debatable that this process would qualify as work.

Page 31. Figure 6. Here and in the text, it is implied that the N6 -NH2 group interacts with Fe3+. In A-T base pairing N6 interacts with the O atom from the carbonyl group of T through a polarized delta+ proton. What would be the driving force in its postulated interaction with Fe3+? In b, the metal ion is hexacoordinated but the coordination geometry is not clear. In any case, it doesn't look octahedral. Maybe the authors should do some modeling to check whether the proposed arrangement is plausible. An experimental confirmation of their model would be even better.

Reviewer #2:

Pinna et al. present a study on the prebiotic formation of ATP from the Fe3+ catalysed reaction between ADP and acetyl phosphate. The authors attempt to answer a very important question in prebiotic chemistry which is how did ATP become the principle energy currency in Life and, in particular, why did ATP become dominant over the other nucleotide triphosphates (NTPs) for this purpose.

The authors extend the work of Kitani (Ref 32 and 33) who first demonstrate that ATP could be formed from a reaction between acetyl phosphate (AcP) and ADP in the presence of Fe3+ ions. The authors explore whether other metal ions can also catalyse this reaction but find that Fe3+ is unique in its ability to do so. The authors further explore the effect of pH, temperature, salt solutes of varying concentration and pressure on the reaction. The authors repeat the pH experiments carried out by Kitani (Ref 33) which affirm the reaction proceeds optimally at mildly acidic pHs. The temperature optimum was at 30 oC. Higher salt concentrations significantly affected the reaction and pressure had a negligible effect. The authors then proceed to test different prebiotically plausible phosphorylating reagents to see if there are others apart from AcP that can perform the reaction. The authors find that carbamoyl phosphate (CP) is the only other phosphorylating reagent which works. Next the authors explore whether Fe3+ catalysed phosphorylation of ADP by AcP also works for other NTPs. The authors find that the reaction exclusively phosphorylates ADP. Finally the authors attempt to unravel the mechanism behind why only ADP is phosphorylated. The authors attempt the phosphorylation of pyrophosphate to determine whether the nucleobase is important and conclude that it is. The authors conclude that a purine ring is important due to the lack of phosphorylation of the pyrimidines. Of the three purine rings explored only adenosine showed phosphorylation and therefore the authors conclude that the N6 amine is essential as both guanosine and inosine have a carbonyl group at the 6-position on the purine ring. The authors then determine that the optimal stoichiometry for ADP and Fe3+ is 1:1. Following Kitani they also find that addition of Mg2+ or Ca2+ increases the yield of the reaction. Lastly the authors conduct a MALDI-ToF experiment to ascertain whether Fe3+-ADP complex stacking is present in the reaction and conclude that it isn't. This all leads to propose the mechanism in Figure 6 whereby a macro chelate complex is formed between a single Fe3+ and a single ADP and the adenosine coordinates to the Fe3+ via the N6-amine and the pyrophosphate. AcP then coordinates to the Fe3+ and the phosphate is transferred and then Mg2+ displaces Fe3+ to promote catalyst turnover. Given that only ADP is phosphorylated to ATP the authors therefore conclude that this is the reason why Life uses ATP as the universal energy currency and that this has its origins in prebiotic chemistry.

I consider that the key result in this paper, namely that only ATP is phosphorylated and not the other NTPs, to be a very interesting result and this will be of great interest to the Origins of Life community. The paper is overall well written and well referenced. The experiments up to and including the demonstration that only ADP is phosphorylated by the reaction have been performed well, although I believe an additional control experiment is required to prove that ADP is being phosphorylated only by AcP in the presence of Fe3+. However, I have a number of serious concerns with both the experiments used to probe the mechanism of the reaction (the 31P NMR spectroscopy and the MALDI ToF experiments) and with the mechanistic interpretation of the results from these experiments which run contrary to what is known in the literature. The authors need to address these concerns in detail. Furthermore, the final conclusion of the paper where the authors claim that they have solved the reason why ATP is the universal energy currency of Life is not fully substantiated by the results presented in the paper. The authors have significantly overstretched themselves in making this claim and they need to tone down the conclusion of the paper substantially. I detail my reasoning behind this assessment in comments below.

[Page 6 Line 146] 

In the Figure 2 caption the authors' provide rate constants for the reaction at different pHs and temperatures but no detail is provided as to what rate equation they used to determine these rate constants. Are these rate constants experimental first order rate constants?

[Page 7 Line 183]

The authors perform a control experiment whereby the reaction was performed in the absence of Fe3+ catalyst (Figure 2d). No ATP was detected in this control experiment which demonstrates that the formation of ATP requires Fe3+. 

However, in order to prove the reaction requires acetyl phosphate an additional control experiment is required whereby the reaction is performed in the presence of Fe3+ but no acetyl phosphate. The reason this control experiment is required is because it is also possible that ATP could be produced from the reaction between two ADP molecules, i.e. ADP + ADP -> ATP + AMP, and this reaction could be catalysed by Fe3+. This reaction is known in the literature, see W. Huck Nat. Commun. 2021, 12, 5517.

It is notable that in the HPLC chromatogram in Figure 1b that a significant peak of AMP is also present. This could indicate that the ATP (at least partially) formed via a 'disproportionation' reaction between two ADP molecules.

The authors need to perform this control experiment to confirm whether or not acetyl phosphate is fully or only partially responsible for the phosphorylation of ADP.

[Page 8 Line 197] and [Page 14 Line 426]

The authors demonstrate that, in addition to acetyl phosphate, carbamoyl phosphate (CP) can also phosphorylate ADP to ATP in the presence of Fe3+ albeit in a lower yield than for AcP. This is a very nice result as it demonstrates that there are multiple prebiotic routes to the formation of ATP with Fe3+. 

The authors correctly attribute the lower yields of ATP with CP to the decomposition of CP into Pi and cyanate. The citation of Jones and Lipmann (Ref 57 and Ref 74) is well received as they demonstrated that carbamoyl phosphate is formed in equilibrium with cyanate and Pi. 

Given this good result with carbamoyl phosphate I am genuinely surprised at how dismissive the authors are of the importance of cyanate in prebiotic chemistry when it comes to the Discussion. The easiest prebiotically plausible route to carbamoyl phosphate is through a reaction between cyanate and Pi. While I appreciate that in extant metabolism carbamoyl phosphate is formed primarily through a glutamine deamination pathway which releases ammonia which then reacts with carboxy phosphate to form carbamoyl phosphate, this set of reactions is significantly less facile on the prebiotic Earth than the reaction with cyanate. 

Furthermore the authors state [Page 15 Line 431]: 'That was important as it showed that biologically relevant condensations are possible in water, but differed from modern biochemistry in that cyanate does not feature in extant metabolism.'

This is not correct. Cyanate does feature in extant metabolism. As already mentioned above, and indeed cited by the authors, Jones and Lipmann (Ref 57 and Ref 74) demonstrated that in solution carbamoyl phosphate is in equilibrium with cyanate and Pi and therefore cyanate will be present in cells. 

Ultimately, I believe that this dismissal of cyanate weakens the paper and undermines the authors' own promising result that Fe3+ catalyses the phosphorylation of ADP by carbamoyl phosphate. I consider that the demonstration of multiple pathways to the formation of ATP to be a strength of the paper and not a weakness.

One experiment the authors could try to raise the yield of ATP with carbamoyl phosphate is to perform the reaction in the presence of excess cyanate which will shift the equilibrium between CP and cyanate + Pi towards carbamoyl phosphate formation. Such a strategy was used in recent publication whereby excess cyanate enabled carbamoyl phosphate formation which in turn facilitated the formation of a phosphoramidate imidazole phosphate that was also demonstrated to phosphorylate ADP to ATP (W. Huck Nat. Commun. 2021, 12, 5517). A possible downside to the addition of excess cyanate here could be that coordination of cyanate to the Fe3+ could interfere with its ability to act as a catalyst. Nevertheless, I think it would be interesting reaction to try. 

Another interesting set of reactions to try would be whether the Fe3+ catalysed CP phosphorylation also works with the other NDPs or whether it only works with ADP.

[Page 9, Line 230]

One strand of evidence that the authors claim for importance of the nucleoside moiety in the formation of ATP is that they perform an experiment with pyrophosphate and do not observe triphosphate formation. They analyse this reaction via 31P NMR spectroscopy and the results are shown in Figure 4f and SI Figure 4. The authors note that this analysis is complicated by the paramagnetism of Fe3+ which can reduce signals in the 31P NMR spectra but nevertheless describe this effect as 'cosmetic and does not conceal the absence of triphosphate in the reaction mixture'.

I have significant concerns with Figure 4f and SI Figure 4. In particular, I am concerned with the result for the reaction of acetyl phosphate with pyrophosphate in the presence of Fe3+ as from the 31P NMR spectra (SI Figure 4, green and purple traces) as no inorganic phosphate Pi is present after 5 h. I find it inconceivable that acetyl phosphate was not hydrolysed during this reaction as acetyl phosphate is shown to hydrolyse significantly in the absence of Fe3+ (SI Figure 4, orange and yellow traces). I am very sceptical that Fe3+ inhibited the hydrolysis reaction - if anything I would expect it to catalyse the hydrolysis. Given the glaring absence of Pi here, I do not accept the authors' description that the effect of Fe3+ is 'cosmetic'. If, for example, the Pi was lost during the solid-phase extractions then could this not also have happened to any triphosphate formed during the reaction? In summary, SI Figure 4 does not substantiate the authors claim that the adenosine is necessary to produce phosphorylation of a diphosphate.

I believe that the authors are correct in their assertion that Fe3+ does coordinate to the adenosine as it is well evidenced in the literature that metal ions coordinate to the adenosine nucleobase (H. Sigel Chem. Soc. Rev., 1993, 22, 255-267) but that the experiment performed here is inadequate to prove this. 

Instead I would suggest that the authors consider using EPR spectroscopy measurements on the NDP-Fe3+ complexes to gain mechanistic insight into their very interesting results. EPR spectroscopy can be performed upon solutions and in conjunction with electronic structure modelling studies can be used to give mechanistic detail. As the authors are in the U.K. I would suggest contacting at Professor Victor Chechik at the University of York who I know has extensive experience in EPR spectroscopy and mechanistic studies.

[Page 9 Line 243] and [Page 10 Line 273] and [Page 11 Line 306]

The authors attribute only observing phosphorylation of ADP and not upon other NDPs to the coordination of the N-6 on adenosine to Fe3+ while other nucleobases lack an N-6 amine and therefore are unable to coordinate to Fe3+. This leads the authors to propose the formation of macro chelate complex between the N-6 on adenosine, Fe3+ and the pyrophosphate (Figure 6).

I strongly disagree with the coordination between the N-6 and Fe3+ and therefore I believe their proposed macro chelate complex structure in Figure 6 is incorrect. There is little to no evidence in the literature that the N-6 position of adenosine participates in the formation of a macro chelate complex that the authors describe

Coordination chemistry dictates that the N-6 amine on adenosine should be the second least able nitrogen on adenosine to coordinate to Fe3+ (after the N-9 which is bonded to the ribose). This is because the lone pair on the N-6 position can participate in the aromatic system on the adenosine ring and is therefore poorly available to coordinate. The lone pairs on N-1, N-3 and N-7 are orthogonal to the aromatic ring and therefore can effectively coordinate to metal centres.

Numerous studies have shown that in solution a macro chelate complex is formed between N-7 on adenosine, a metal ion and the pyrophosphate and not the N-6 position. See H. Sigel Chem. Soc. Rev., 1993, 22, 255-267 and references therein for an overview of this topic. For specific evidence of ADP-N-7 macro chelate complex with a metal ion, see H. Sigel J. Am. Chem. Soc. 1983, 105, 5891-5900. This paper also shows that a lower percentage of the M2+ ADP macro chelate complex is in the closed form than for GDP and IDP complexes (see Table V).

The authors dismiss the importance of the N-7 macro chelate complex to the phosphorylation chemistry stating [Page 9 Line 247]: 'We infer that the critical moiety in the adenosine ring for phosphorylation by AcP with Fe3+ as catalyst must be the N6-amino group of adenosine, as the IDP and GDP ring structures are equivalent elsewhere. In particular, from a mechanistic point of view, we note that the N7 is equivalent in all three purine rings, so although this might also interact with Fe3+, as suggested by others [60-63], it cannot be the critical moiety.'

However, the N-7 position on all three purine rings of adenosine, inosine and guanosine are not equivalent from a mechanistic point of view. In their thinking the authors have neglected to consider that replacing the amine with a carbonyl will affect the properties of other N's in the purine ring. Sigel has demonstrated that the basicity of the N-7 position varies significantly as the pKa of the N7-H+ conjugate acid is pKa(Adenosine) = - 0.2, pKa(Inosine) = 1.06 and pKa(Guanosine) = 2.11 (H. Sigel J. Am. Chem. Soc. 1994, 116, 7, 2958-2971). [While this study focused on NMPs I would expect a similar trend in the basicity of the N-7 position for NDPs and NTPs.] Note that this is a log scale so there is a 100-fold difference in the basicity between adenosine and guanosine. The pKa of the N-7 position has a significant effect on the formation of the macro chelate complex with higher basicities leading to higher proportions of the metal ion nucleotide complex being in the macro chelate form (see Figure 4 in H. Sigel J. Am. Chem. Soc. 1994, 116, 7, 2958-2971).

The differences in basicity between the N-7 positions on the purine nucleobases will affect the strength of the coordinate bond (and possibly the extent of backbonding between the metal centre and the aromatic system on the purine) which will therefore affect both the energies and spatial size of the orbitals on the Fe3+. The ability of Fe3+ ADP macro chelate complex to catalyse the phosphorylation of ADP will depend on its ability to stabilise the transition state. I would suggest to the authors that the reason they only observe phosphorylation on ADP and not GDP or IDP is because of these aforementioned differences in the Fe3+ orbitals that happen to be just right to stabilise the phosphorylation transition state with acetyl phosphate.

The lack of phosphorylation on CDP and UDP here can be attributed to CDP and UDP (as well as their NMPs and NTPs) not forming macro chelate complexes with metal ions and the only interaction in these cases is between the phosphate and the metal ion (see Sigel Chem. Soc. Rev., 1993, 22, 255-267). This is explained in part by the nucleobase more favourably adopting an anti conformation (as opposed to the syn conformation) in relation to the ribose which results in the N-3 facing away from the phosphate/pyrophosphate/triphosphate. 

Later on in the Discussion [Page 11 Line 369], the authors cite a Mossbauer spectroscopy study for evidence of N-6 binding of Fe3+ to ADP (Ref 60 I. Rabinowitz J Am Chem Soc. 1966, 88, 4346-4354). I do not believe that this study gives adequate evidence for N-6 binding. First, this study was conducted in 1966 and all later studies contradicted the finding of N-6 binding and have shown that the binding is instead to N-7. Secondly, Rabinowitz et al. make the same mistake as the authors in comparing the results from inosine to adenosine to justify the choice of the N-6 position without considering how replacing the amine at the 6-position with a carbonyl will affect the basicity of other N in the purine nucleobase. Thirdly, Mossbauer spectroscopy is conducted on solid samples (in the Rabinowitz paper the pH of the sample was adjusted and then lyophilised to a powder). The change from solution to solid state means that the results are less reliable than the techniques in later studies which analysed the structure of the metal ion ADP complexes in solution.

In summary, I believe the weight of evidence is that an ADP N-7 macro chelate complex is formed in solution and not the N-6 macro chelate complex and consequently the authors need to adjust their text and Figure 6 to reflect this.

[Page 9 Line 257]

The authors show that the addition of low Mg2+ and Ca2+ concentrations to the reaction leads to higher yields of ATP. The authors justify this increased yield by stating that the Mg2+ and Ca2+ can stabilise the ATP to hydrolysis which is the explanation also provided by Kitani for this phenomenon. However, the authors go further and claim that the addition of Mg2+ and Ca2+ promote catalyst turnover by liberating the Fe3+. This would suggest that the rate determining step for the reaction is the liberation of the Fe3+. I am sceptical of this claim as I would have thought the phosphate transfer from AcP to ADP would be the more likely rate determining step. I would favour the explanation that the Mg2+ and Ca2+ stabilise ATP to hydrolysis. If the authors wish to prove that Mg2+ and Ca2+ are present in the rate determining step in the catalytic cycle then they would need to demonstrate that the reaction order is dependent on these ions. 

An alternative explanation for the dependence on Mg2+ and Ca2+ is that they participate in the formation of a dimeric complex (e.g. FeMg[ADP2]) which is also catalytically active. 

[Page 10 Line 261]

The authors make an assessment of whether Fe3+ is acting as a catalyst by measuring the rate of the reaction with different concentrations of the ADP substrate and then apply a Michalis Menten fitting to the data (Figure 5b). In these experiments the authors use [Fe3+] = 0.5 mM and they don't state the concentration of AcP but I presume it is 4 mM. The ADP concentration is then varied between approx. 0.1 mM - 4 mM. In these circumstances the application of a Michalis Menten kinetic analysis is technically speaking not appropriate as the Michalis Menten kinetic analysis assumes that the substrate concentration is in significant excess over the catalyst concentration (a 10-fold excess of substrate is usually regarded as the minimum viable). This is because the Michalis Menten kinetic analysis assumes pre-equilibrium formation of the enzyme substrate complex and rate determining kcat. In the experiments performed by the authors the concentration of ADP is either lower than Fe3+ or at most in an 8-fold excess. Thus, this Michalis Menten fitting is strictly speaking not valid although I do accept that the level off is indicative of catalysis. 

Like for Figure 1, insufficient detail is provided by the authors as to how they measured the rates of reaction here. I appreciate the effort the authors have gone to in order to measure the rate of reaction and this makes a good addition to the study but I would like to see the kinetic data and equation used to determine the rate constant for the data in Figure 5b put into the Supplementary Information so that I can properly assess this aspect of the study. 

[Page 10 Line 267] and [Page 11 Line 313]

The authors claim that a dimeric structure of ADP with Fe3+ (SI Figure 6) is not the catalytically competent species based upon a MALDI-TOF analysis which did not show the stacked species.

I would dismiss the validity of using MALDI-TOF to assess this claim on two criteria.

First, the MALDI-TOF is performed on a solid sample in a matrix and not upon a solution and thus the species present may be altered by the change in state.

Second, the MALDI analysis requires sample preparation steps that could have significantly interfered with this stacked dimer species. The main drive force behind why the stacking occurs in solution will be to minimise the hydrophobic surface area presented to the solvent by the nucleobases. During the MALDI sample prep the sample is washed with a 50:50 acetonitrile:water solution and acetonitrile is a significantly non-polar solvent it could disrupt the hydrophobic stacking and thereby break this stacked dimer apart. 

I think the authors should remove this MALDI study from the paper as it is not possible that this experiment can give any effective insight into whether a dimer is formed during the reaction in solution.

Furthermore, the evidence from the literature that metal complexes can form dimeric structures (with intermolecular adenosine N-7 metal ion interactions) and can also stack is fairly substantial, see H. Sigel J. Am. Chem. Soc. 1983, 105, 5891-5900. I would therefore not dismiss the possibility that the catalytically competent species for the phosphorylation reaction is the dimeric form. Note too that the optimal 1: 1 stoichiometry for the reaction applies to both the monomeric complex (1:1) and the dimeric complex (2:2) and therefore cannot be used to distinguish between these mechanisms. 

[Page 11 Line 301]

As acknowledged by the authors, there is not a link between Fe3+ and the phosphorylation of ADP to ATP in biology. I would agree with this as to the best of my knowledge ATP synthesising enzymes do not make use of Fe3+ as a cofactor. For example, the acetate kinases in Ref 41 use Mg2+ as a cofactor. The authors advocate a position on the origins of life whereby the chemistry out of which life emerges should align as closely as possible to the biochemistry of extant life. If Fe3+ were to play such a crucial role at the origins of life then why would this not have been retained in biology? Such a question is raised by the authors but they do not provide any adequate answer to it.

[Page 11 Line 296] and [Page 15 Line 447]

At the opening of the Discussion the authors state that: 'Taken together, these findings suggest that the pre-eminence of ATP in biology has its roots in aqueous prebiotic chemistry. … This implies that ATP became the universal energy currency of life not as the endpoint of genetic selection or some frozen accident, but for fundamental chemical reasons, and probably in a monomer world before the polymerization of RNA, DNA and proteins.'

And then end the Discussion stating: 'If so, then ATP became established as the universal energy currency for reasons of prebiotic chemistry, in a monomer world before the emergence of genetically encoded macromolecular engines.'

The authors claim that their results justify why ATP became the universal energy currency for life and moreover did so before the advent of enzyme catalysts. This is a bold claim and unfortunately the authors have significantly overextend themselves in making it as such a claim cannot be fully supported by their results. 

My main objection here is that the authors have only demonstrated half of what is necessary to solve why ATP became Life's universal energy currency. To justify why ATP became Life's energy universal currency at the origins of Life requires consideration of both how ATP could have formed and how ATP came to be not just the dominant phosphorylating NTP but also the dominant phosphorylating reagent at the origins of Life. 

The authors' results do provide a plausible explanation for why the formation of ATP became dominant over the other NTPs but this is as far as the authors can reasonably take their conclusions based upon their results. 

The huge oversight by the authors here is that they do not seriously consider how ATP performed prebiotic phosphorylations. This receives scant attention in the paper and is only addressed in the penultimate sentence. The authors state [Page 15 Line 443]: 'Once formed, ATP would promote intermediary metabolism through phosphorylation'

This statement trivialises an enormous challenge in prebiotic chemistry which is how to perform prebiotic phosphorylation reactions. For example, a central challenge to prebiotic phosphorylations is how to overcome the high activity of water which results in the hydrolysis of the phosphorylating reagent rather than transfer of phosphate to an organic molecule. The authors show that solutions with a high water activity favour formation of ATP [Page 6 Line 166], however such conditions will be the least effective for the phosphorylation of organic compounds meaning that the authors will almost certainly need a very different set of conditions for the phosphorylation by ATP. Under these different conditions would ATP necessarily be a better phosphorylating reagent than the other NTPs? Furthermore, I would argue that ATP is a very poor choice as a prebiotic phosphorylating reagent. This is because ATP is a very stable compound that has a hydrolysis half-life of around 2 years at pH 8 and 25 oC in synthetic sea water (H. Hullet Nature 1970, 225, 1248). Given the high stability of ATP I would not expect its use in prebiotic phosphorylations until after the advent of macromolecular engines e.g. a ribozomyl or enzyme catalyst.

If the authors wish to demonstrate why ATP became Life's universal energy currency then they must consider questions such as the following and address these experimentally: 

Why did life settle upon NTPs to perform reactions in prebiotic chemistry? 

Why did ATP become dominant phosphorylating reagent over the other NTPs? 

How did phosphorylation of organic compounds by ATP overcome the high activity of water?

If, as the authors show, ATP can be selectively formed be a metal ion catalysed reaction could it not also perform metal ion catalysed phosphorylations that other NTPs cannot?

Given the stability of ATP/NTPs Is it not possible that ATP is chosen 'late' in the origins of Life after the origin of macromolecular engines?

The experimental results presented in this paper do not address these questions and therefore do not tackle how ATP could perform prebiotic phosphorylations and thus for the authors to claim that their results show why ATP became Life's universal energy currency is not justified. The authors need to constrain their conclusions to what their experimental results can actually prove.

The authors must also consider how prebiotic chemistry also produced the other NTPs as Life has settled on the use of NTPs for synthesis of DNA and RNA. This requires an alternative route(s) to NTPs. If this alternative route(s) to NTPs produced all NTPs at a higher rate than the Fe3+ catalysed phosphorylation of ADP by acetyl phosphate then this would undermine the authors' case that ATP was formed in excess over other NTPs.

Minor considerations:

In the caption of Figure 4 it would be better to call it a nucleobase rather than base.

For Table 1, the authors may wish to cite a recent paper on the geological presence of cyclic trimetaphosphate: Britvin, S. N. et al. Cyclophosphates, a new class of native phosphorus compounds, and some insights into prebiotic phosphorylation on early Earth. Geology 49, 382-386 (2020) https://doi.org/10.1130/G48203.1

[Page 8 Line 207] with regards to cyanate being a condensing agent, in addition to ref 58 the authros may wish to also cite R. Pascal J. Am. Chem. Soc. 2006, 128, 7412-7413.

Mathew Pasek wrote a very good recent review of prebiotic phosphate chemistry that the authors may wish to cite: Chem. Rev. 2020, 120, 11, 4690-4706 10.1021/acs.chemrev.9b00492

Reviewer #3:

This is an interesting paper which relates to a chemical, non-enzymatic synthesis of ATP from ADP and Pi using acetyl phosphate and ferric ion. From a chemical perspective alone, the work is not necessarily original but it is more in the context of prebiotic chemistry and abiogenesis that the work is couched and consequently, it has been reviewed here in that context.

It is pertinent to comment first on the authors contextual history in the field which is significant. The authors have a long-standing presence in the sphere of prebiotic chemistry, specifically linked to the role of concentration and charge gradients as a source of energy to drive the emergence of biology.

The present contribution centres on the conversion of adenosine diphosphate (ADP) and inorganic phosphate (Pi) to adenosine triphosphate (ATP) using acetyl phosphate and ferric ion as co-factors, without the need to invoke multi-subunit protein complexes as catalysts. There a number of intriguing aspects to this work, including (i) the specificity of acetyl phosphate in this process (although carbamoyl phosphate also shows some activity) and (ii) the specificity of ferric ion. Both of these are sufficiently intriguing as to ask why these components appear to be so unique and what is the level of significance in this from a pre-biotic perspective.

From a practical methods perspective, these looks perfectly sound to me, chemically and analytically. Technically, relatively little discussion is devoted to the kinetic aspects of the study but the focus is more on thermodynamics. From a prebiotic energetic context this is certainly of value, however, within the context of dynamic kinetic behaviour as in biology, it tells only a part of the story. 

In terms of comments, I have two to pass on to the authors for their consideration: (i) it is clear that the authors place great emphasis on the specificity of both acetyl phosphate and ferric ions in this chemistry, it is central to the actual conclusions. Figure 6 proposes a potential mechanism for how this might work. However, what is rather surprising missing is a more thorough molecular modelling analysis of just how reasonable this actually is. This is a significant omission and one that could logically be quite straight-forward to add. I would not suggest full molecular dynamics calculations be necessary at his stage but simple bench-top molecular mechanics modelling should be included. (ii) The second point is rather more contextual, but goes to the central importance of the present chemistry in primitive energy transduction. The authors emphasise the value of ATP as a primitive energetic "battery". This has some validity of course, but I feel it really does need to be nuanced a little more for the wider and especially biological audience. For sure, ATP can act as an energetic battery in terms of direct phosphorylation. However, the situation becomes more complex when it comes to linking ATP/ADP to proton gradient or indeed to enzymatic phosphorylation-dephosphorylation. The energy which results from ATP hydrolysis in biological systems is transduced via multi-subunit complex enzymes and converted to mechanical energy through a rotatory mechanism. Because the actual hydrolysis reaction of ATP is enzymatically very fast (femtosecond), it is generally believed that this chemical hydrolysis step itself is NOT the step in the whole process where energy is transduced, but rather the binding-de-binding of ADP/ATP components from the enzyme binding sites themselves. This is what allows the ATP hydrolysis to be connected to proton gradients. Non of this is made clear in the current analysis which makes it a little less clear just how significant pre-enzymatic energy transduction using ATP might have been and what could have acted as a pre-enzymatic catalyst and then also, how was that energy transduced from ATP to a gradient or other usable energy source. With a little more nuance, the authors might be able to place their work in a slightly more realistic context within the pre-biotic milieu.

Reviewer #4:

In "A prebiotic basis for ATP as the universal energy currency" the authors present an intriguing argument that the origin of ATP as the dominant energy carrier for prebiotic chemistry resides in the fact that acetylphosphate can phosphorylate ADP to form ATP (and ADP alone!) in the presence of catalytic Fe3+. Indeed, the authors demonstrate that all three of these components are uniquely situated to perform this task: changing any nucleobase, metal catalyst, or phosphorylating agent and the yields plummet. Given the specificity of adenosine in this task, the authors argue that ATP was predisposed to be the dominant energy currency of life.

In general, the experiments are well-thought out and easy to follow. The analytical tools appear reasonable, within the limitations of these instruments. The reasoning within the discussion is sound enough (in general) for the experiments that have been carried out.

I have several concerns regarding the conclusion, other phosphorylation agents that were not discussed in the text, the prebiotic geochemistry of Fe3+, and some of the methodology and reporting of results. These are explored below.

1. The main point of the paper is that ADP to ATP occurs readily only by phosphorylation by acetylphosphate with catalytic Fe3+, under a specific set of aqueous conditions (acidic pH, T <= 50°C, lacking too much salt or divalent cations). Thereby, the conclusion is that ATP as the dominant is not a "frozen accident" but is a consequence of the environment in which life originated. However, such a conclusion appears to be a circular argument: the specificity of conditions needed to generate ATP from ADP means that ATP was specifically chosen because of those conditions. If the conditions are not met on the prebiotic earth (e.g., no Fe3+, no acetylphosphate), then no ATP would be specific. There are other routes to generating prebiotic NTPs (detailed in points 2+3). If ATP is a "frozen accident" then these routes would be just as plausible. To some extent, this point is allayed by the discussion on acetylphosphate, as acetylphosphate is present in modern biology. However, if we presume the earth was somehow "different" with respect to its prebiotic geochemistry (see Benner et al. ChemSystemsChem 2020), and thereby other reactants not observed in geochemistry or biochemistry today would have been plausible (e.g., cyanate as discussed by the authors), then prebiotic chemistry need not follow modern biochemistry.

Prebiotic chemistry is often OK with "frozen accidents" as well, as the chirality of biomolecules is generally assumed to be a consequence of the possibly arbitrary higher abundance of molecules with the biological handedness. Also, some aspects of the genetic code (while highly evolved) were likely randomly frozen in as well. 

2. The paper is lacking discussion of the amidophosphates, which have been explore by the Krishnamurthy lab for their ability to phosphorylate organics. Gibard et al. (Nat Chem 2018) demonstrated phosphorylation of NDPs to NTPs using diamidophosphate. Recently, Lin et al. (Ange. Chem 2021, very recently published, but building on Gibard et al. 2018) demonstrated NMPs are transformed into NTPs via reaction with diamidophosphate. Amidophosphates have a parallel in modern biochemistry in forms such as creatine phosphate.

3. Triphosphorylation of nucleosides via cyclic trimetaphosphate and a ribozyme has been reported by Akoopie et al. (Sci Adv 2021) and presents an alternative explanation for NTPs: post-RNA world (Moretti and Muller 2014). Although trimetaphosphate has its own issues with prebiotic provenance (e.g., Keefe and Miller 1995), some routes have been shown to these compounds (Yamagata et al. Nature 1991, Pasek et al. ACIE 2008). Kim and Benner (Astrobiology 2021) also showed NTPs from trimetaphosphate.

4. The prebiotic geochemistry and aqueous chemistry of Fe3+ is not adequately considered. Fe3+ is generally soluble only at low pH, well below the pH used by the authors. Iron redox diagrams (for instance, see Takeno 2005, Atlas of Eh-pH diagrams, Figure 47) show that soluble Fe3+ exists only at pH <2.3. Above a pH 2.3, Fe3+ reacts with water to produce Fe(OH) 2+, and then Fe(OH)2 +. These are not Fe3+. This chemistry is roughly reflected in the results presented by the authors. At the higher pH, Fe3+ precipitates out as Fe(OH)3 or FeOOH. The drop in yield at these higher pH values likely corresponds to a precipitation of Fe3+ or the production of neutral Fe species (such as Fe(OH)3 and FeOOH). The actual aqueous ionic chemistry of Fe3+ at the conditions being explored (Figure 6) is likely completely different than those being discussed here as Fe3+ (aq) isn't Fe3+ at a pH of 5.5.

To this end, it's also unclear if the pH of the solution was adjusted before or after the addition of Fe3+ (as Fe2(SO4)3). I presume after, but if before, Fe3+ is an acidic cation that will drive the pH to a lower value.

Furthermore, Fe3+ requires highly oxidizing conditions at low pH, well above the point of the H2S/SO42- redox couple (a full 0.6 V at pH 2), and given that thioesters are discussed in the context of this prebiotic chemistry, these highly disparate redox conditions themselves would introduce disequilibrium. The arguments for Fe3+ could also be expanded in the text, as Fe3+ also forms from the reaction of Fe2+ with H2O2, the latter formed by the photolysis of water (via Fenton chemistry). 

In addition, Fe3+ is unstable in non-acidic water, eventually precipitating out as hematite (Fe2O3). The reaction 2Fe3+ + 3H2O = Fe2O3 + 6H+ has a reaction K of around 0.7 at 30°C, which means that at a pH of 5.5., the concentration of Fe3+ should be sparingly small at equilibrium. That's not to say solutions can't be supersaturated with respect to Fe3+, but that high concentrations of Fe3+ should be geologically ephemeral. 

5. The chemistry of iron (III) is more closely replicated by aluminum (III) than by the other species investigated. The ionic size of Al3+ is similar to Fe3+ and the chemistry of these ions in water vs. pH is similar. It may be worth replicating these experiments with Al3+ to see if Fe3+ is still unique in its chemical potential for ADP to ATP, especially in comparison to the species chosen (Co3+, Cr3+, and Mo3+ are all quite different from Fe3+ in aqueous chemistry).

6. The results do not discuss the possibility of precipitates of the various components investigated. Does iron(III) form red FeOOH precipitates? Is there any precipitation of Mg-ADP? Is phosphate being precipitated by the cations? This is also a question of mass balance: Is the amount of phosphorus in ADP and AcP that was added to the beginning of the reaction the same as the total organophosphate that came out? If not, were there precipitates of some of these ions (perhaps ATP in some of the Mg/Ca experiments?)? If no precipitates were noticed, then this should be noted in the text. The Fe3+ should certainly provide precipitates, given the "gumminess" of this ion.

7. The study of the pyrophosphate phosphorylation (figure SI 4) could be improved significantly by changing the solution pH of the NMRs acquired. The spectra are quite broad, as acknowledged by the authors. However, given the multiplicity of the PPPi peak, and its expected low concentration (~20% of ADP was converted in these experiments) would a triphosphate peak even be visible if it was at ~5-10% yield in such conditions? The authors could attempt to precipitate Fe3+ using NaOH or Na2S to sharpen up the spectra and investigate for PPPi. As it is, the current spectra do not provide confidence in either its presence or absence.

COMMENTS FROM THE ACADEMIC EDITOR:

The manuscript by Pinna et al. characterizes a previously described reaction (as the authors note) whereby acetyl phosphate phosphorylates ADP to produce ATP, catalysed by Fe3+. However, the authors then go on to demonstrate the specificity of the reaction for ADP over other ribonucleotides. That result on its own makes the work publishable (generally speaking). The manuscript is written eloquently, and the message of the authors is clearly expressed. However, there are serious flaws in the interpretation of the data and in the placement of the data in a broader context. Therefore, in my opinion, the manuscript would need to be substantially revised for consideration.

All of the reviewers gave constructive criticisms that should be considered. Reviewer 2 provides a longer, extensive list of criticisms. I do not find the requests unreasonable, as several of the points seek to clarify and/or correct shortcomings. However, I would not push for the EPR analysis to be performed. EPR would be insightful, but that would take much time, and most of the criticisms can be addressed without EPR.

As the reviewers have done a thorough job, I will only briefly comment on a few of the points raised. The complex described in Figure 6 is questionable. If this were to be taken more seriously, then some type of modelling and additional experiments would be needed. As reviewers 1, 2 and 3 noted, and as is well known in coordination chemistry, the coordination of metal ions to the N6 position of adenosine is not reasonable since the electrons of N6 are distributed across the aromatic ring. Even when deprotonated at high pH, this position is a poor ligand.

It is also necessary to run a control demonstrating that ADP + ADP � ATP + AMP is not occurring, particularly since this reaction has been documented before (as reviewer 2 notes).

As reviewers 1 notes, the data are consistent with a “surface pond or lake” rather than a hydrothermal vent. It would be better if the authors let their own data guide them rather than to force interpretations onto previously espoused convictions. I can understand if the authors do not want to advocate for surface pond or lake conditions, but a more fair and balanced interpretation of the data are warranted and would contribute well to the field.

Regarding more fresh water conditions, on pages 4-5 the manuscript states “But the fact that substrate-level phosphorylation of ADP to ATP can be accomplished by AcP in water says nothing about whether his mechanism actually holds prebiotic relevance.” This is a fair point, but the conditions used by Pinna et al. are not necessarily anymore prebiotically plausible. Basically, extremely low levels of salt are necessary to allow for the reaction to occur. The fact that NaCl and MgCl2 greatly inhibits the reaction is alarming in terms of prebiotic relevance, especially if you are a proponent of hydrothermal vents. The fact that 1-2 mM Mg2+ promotes the reaction does not strengthen the arguments. Environmental concentrations of Mg2+ are much higher. Even the concentration inside of a cell is much higher. Unless one were to invoke freshwater conditions, the inhibition by Mg2+ (and Na+) suggests that the data are not relevant to prebiotic chemistry, as it is difficult to imagine how such chemistry could have been mediated on the prebiotic earth.

To help the reader, it would be good to indicate what pressures are typically found in hydrothermal vents. That way, the reader would understand what kind of environments the 80 bar data correspond to.

As reviewer 4 notes, it is alarming that some highly relevant work was overlooked, including work from the Huck group (Maguire et al. Nat Commun 2021, 12, 5517) and the Krishnamurthy group (Gibard et al. Nat Chem 10, 212–217). These are extremely relevant studies that explore prebiotic phosphorylation. If a fair and balanced perspective were to be taken, then the implications of such work should be taken into consideration.

Similarly, the neglect of the potential importance of cyanate further suggests biases. This is even more evident since the dismissal of cyanate is inconsistent with how the authors seem to view prebiotic chemistry, in that things that are not found in biology are not considered. Using the same logic, we should dismiss a role of Fe3+ in prebiotic phosphorylation, since that is not found in biology.

Reviewer 4 raises several points regarding the solubility and availability of Fe3+. Additionally, the photooxidation of Fe2+ to Fe3+ has been experimentally demonstrated (Bonfio et al. Nat Chem 9, 1229–1234).

Finally, as reviewer 1 notes, the speculation that ATP would be synthesized at acidic pH and then enter a protocell with an internal alkaline pH does not appear logical. Nucleotide triphosphates are not permeable across membranes that are thought to be prebiotically plausible. If membranes were stable to high concentrations of Mg2+, then perhaps enough Mg2+ would be bound to neutralize the charge, but such concentrations of Mg2+ would inhibit phosphorylation of ADP to ATP. It is difficult to imagine a scenario in which the gradients that the authors wish to invoke were established.

There is something interesting in the submitted manuscript. The specificity for ADP is new and definitely interesting. The details of the chemistry need to be improved, and a fairer representation of the implications of the work and how that fits into prior studies is needed.

---

## [Decision Letter · Decision Letter 2]

5 Aug 2022

Dear Nick,

Thank you for your patience while we considered your revised manuscript "A prebiotic basis for ATP as the universal energy currency" for publication as a Research Article at PLOS Biology. This revised version of your manuscript has been evaluated by the PLOS Biology editors, the Academic Editor, and three of the original reviewers. As intimated in my previous email, the original Academic Editor was not able to continue handling your paper, so we have assigned a new Academic Editor to help us with these final stages.

Based on the reviews, we are likely to accept this manuscript for publication, provided you satisfactorily address the remaining points raised by the reviewers. Please also make sure to address the following data and other policy-related requests.

IMPORTANT:

a) Please address the remaining concerns from the reviewers.

b) Just in case this is helpful for formulating your revisions, when I discussed the reviewers' comments with the Academic Editor, they said "R1 raises important clarification points on the charge of ADP, and how purine nucleotide synthesis could still be reasonably still occur on minerals presuming, for instance, the presence of nucleosides. The presence of nucleosides has been investigated for a substantial period of time, and though biologically it requires a rather large quantity of metabolic energy, there’s been a long history of thinking about nucleotide formation abiotically. From Oro’s early experiments showing adenine synthesis from HCN, to the formation of ribose from formaldehyde, to condensation and phosphorylation, there have been several routes to making and assembling nucleotides that do not require biology that have come from a prebiotic chemistry perspective. This would be a more heterotrophic view of the origin of life than suggested by the autotrophic view of the authors, and both should be viewed as reasonable schools of thought." 

c) Please address my Data Policy requests below; specifically, we need you to supply the numerical values underlying Figs 1ABC, 2ABCD, 3ABCDEF, 4ABCDEF, 5ABC, 6ABCD, S1AB ,S2, S3, S4AB, S5, S7ABCD, S8, either as a supplementary data file or as a permanent DOI’d deposition like Zenodo, Dryad, Figshare, etc..

d) Please cite the location of the data clearly in all relevant main and supplementary Figure legends, e.g. “The data underlying this Figure can be found in S1 Data.”

We expect to receive your revised manuscript within two weeks. 

*Published Peer Review History*

*Press*

Best wishes,

Roli

Roland Roberts, PhD

Senior Editor,

rroberts@plos.org,

PLOS Biology

DATA POLICY:

Regardless of the method selected, please ensure that you provide the individual numerical values that underlie the summary data displayed in the following figure panels as they are essential for readers to assess your analysis and to reproduce it: Figs 1ABC, 2ABCD, 3ABCDEF, 4ABCDEF, 5ABC, 6ABCD, S1AB ,S2, S3, S4AB, S5, S7ABCD, S8. NOTE: the numerical data provided should include all replicates AND the way in which the plotted mean and errors were derived (it should not present only the mean/average values).

DATA NOT SHOWN?

REVIEWERS' COMMENTS:

Reviewer #1:

[new comments, indicated by "R1':" are interdigitated]

R1: The manuscript by Pinna et al. reports on the detailed characterization of the phosphorylation of ADP by acetyl phosphate (AcP) to form ATP in the presence of ferric ions. This previously observed reaction appears to be surprisingly specific as no other NDPs or transition metal ions can carry out equivalent reactions to any significant extent. ADP phosphorylation appears to be optimal at 30°C, mildly acidic pH and low salt. The authors provide explanations for their observations within the context of their already well-established hypothesis of life's origins in a hydrothermal vent. Besides the fact that, taken face value, the observations reported here would rather favor a surface pond or lake as the cradle of life, some of their propositions need to be further substantiated.

R: We thank R1 for this clear synopsis. We agree that taken at face value our observations might seem to favour freshwater environments and we make this clearer in the revised version (p14 lines 401-408). We have also revised the order of the Discussion to bring up this passage before the section on work, as the paragraphs on work may have obscured this point in the original version. 

R1': OK.

R1: Page 3 "ATP in a prebiotic, monomeric world". This assumes that life originated in bulk water. But, if, instead, life first appeared on mineral surfaces then the needed energy could have been provided through the oxidation of H2 and CO by, for instance, catalytic FeNiS minerals. Only when more evolved protocells liberated themselves from those surfaces, and moved to bulk water, would have ATP become essential.

R: We have revised the Introduction to make our reasoning clear here. We agree with R1 that the earliest forms of metabolism are indeed exergonic and could have occurred on mineral surfaces. But it is unlikely that this could account for purine nucleotide synthesis, for example, which has six phosphorylation steps. We lay out this perspective in the revised Introduction, specifically on pages 3-4, lines 69-83.

R1': In fact, there is no a priori reason to conclude that purine nucleotide synthesis could not have taken place on a mineral surface. The authors should check the paper by Deiana et al. (ChemCatChem 2013, 5, 2832-2834.) who argue that "… the interactions of carboxylate oxygen atoms with surface Ti4+ ions, which act as Lewis acid centers, is expected to withdraw electron density from the C atom, which becomes electrophilic enough to undergo nucleophilic attack by the nitrogen atom of the amine". This is exactly the effect of phosphorylation on the carbonyl carbons of reaction intermediates, which allows for the formation of 9 C-N bonds during purine synthesis. Although acetyl phosphate could have been involved in ATP synthesis other pathways are also possible (see Akouche et al. Angew. Chem. Int. Ed. 2017, 56, 7920-7923 for the remarkable one-pot synthesis of AMP from phosphate, adenine, and ribose on a fumed silica surface).

R1: "Even if only one nucleotide triphosphate can be dominant, the implication of a frozen accident is not a satisfying explanation." Why not? There are different levels of "frozen accidents". This has been discussed by Brandon Carter who, in 1984, proposed that the evolution towards intelligent life required 6 "hard steps", the first one being abiogenesis. Each step was statistically postulated to last between 600 and 800 million years. Although these periods of time may be considered workable in astronomy and geology, in biology they do not make the same sense. In the latter case it is necessary to invoke the occurrence of extremely unlikely (highly contingent) events to explain that time duration. They can be indeed defined as "frozen accidents".

R: We do not agree with R1 about frozen accidents, but this is a relatively trivial point in relation to the paper. We have de-emphasised frozen accidents in the revised Introduction, and instead stressed the centrality of other adenine nucleotide cofactors including NAD, FAD and coenzyme A, all of which point to the importance (and so availability) of adenosine or ATP in early metabolism.

R1': As the authors point out there is no doubt that adenine nucleotide must have been around quite early. A "frozen accident" in this context describes the -most likely- very low global probability for the required reactants to find themselves in the same location under productive conditions. Once that accident happens subsequent evolution will propagate the result in both time and space if it confers an advantage.

R1: Page 13. "Such steep pH gradients could in principle operate across protocells as well as inorganic barriers." The authors propose that ATP would have been synthesized outside protocells, under acidic conditions, to be then transported across a proto-membrane into a cell where, under alkaline conditions, it would have phosphorylated some substrate(s). Have they considered the implications of such a mechanism in terms of ATP/ADP concentrations and diffusion? ATP would have to diffuse towards the protocell and ADP do it in the opposite direction, without risking to disappear into the bulk water. Another crucial point is membrane permeability. As F. H. Westheimer wrote in 1987: "Phosphoric acid is specially adapted for its role in nucleic acids because it can link two nucleotides and still ionize; the resulting negative charge serves both to stabilize the diesters against hydrolysis and to retain the molecules within a lipid membrane. A similar explanation for stability and retention also holds for phosphates that are intermediary metabolites and for phosphates that serve as energy sources." So, one very important role of even proto-membranes should have been to keep charged metabolites inside the cell to sustain metabolism. This, in turn, would have made ADP diffusion out of the protocell difficult to envision.

R: This is a complete misunderstanding of what we had attempted to write, and we can only apologize for not being sufficiently clear. We agree with R1 (and Westheimer's classic paper) about phosphorylated intermediates, including especially triphosphates, not being able to cross cell membranes, even simple fatty acid bilayers. In fact, we envisaged ADP being phosphorylated inside the protocells immediately adjacent to the membrane, subject to proton influx from the acidic exterior. No phosphates ever cross the membrane in our model. We have made this clear in the revised Discussion (page 15, lines 435-443).

R1': I do not think that 'misunderstanding' is the right way to describe the problem in the original text (p 13). There, the authors wrote "…that thin inorganic barriers … can sustain proton gradients…". And later they concluded "… could promote the phosphorylation of ADP to ATP under locally acidic conditions close to the barriers, followed by hydrolysis linked to phosphorylation under more alkaline conditions in the cytosol of protocells". The question is, where are the "locally acidic conditions close to the barriers"? In the cytosol, as well? So, is there an intracellular gradient? This would be quite different from standard biological proton gradients.

The corrected text is fine.

R1: Page 15. "…enabling ATP to drive work even in a prebiotic monomer world." How are the authors defining 'work' in that setting? In contemporary biology work is i) performed mechanically as in muscle contraction and ii) against a gradient of chemical potential in active transport (see Jencks, 1989). This work depends on complex protein-based binding energy modulation and the coupling of intermediate reaction steps. Of course, this kind of work could not have existed prebiotically. At that time the most likely role of ATP (if it already existed) would have been, for instance, phosphorylation to render carbonyl carbons electrophilic enough to be attacked by a nucleophile such as ammonia. It is debatable that this process would qualify as work.

R: Our perspective is grounded in the idea of a monomer world, and we do not envisage any protein-based machinery here at all. We have made this clearer at several points in the revised MS and added a new paragraph to the end of the Discussion (page 15 lines 451- 464) where we discuss specifically what we mean by work in this context (essentially coupling exergonic to endergonic reactions, and in particular the polymerization of nucleotides and amino acids).

R1': OK

R1: Page 31. Figure 6. Here and in the text, it is implied that the N6 -NH2 group interacts with Fe3+. In A-T base pairing N6 interacts with the O atom from the carbonyl group of T through a polarized �+ proton. What would be the driving force in its postulated interaction with Fe3+? In b, the metal ion is hexacoordinated but the coordination geometry is not clear. In any case, it doesn't look octahedral. Maybe the authors should do some modeling to check whether the proposed arrangement is plausible. An experimental confirmation of their model would be even better.

R: We thank R1 for pointing out that base pairing in DNA and agree that our model as originally depicted was incorrect. We have also undertaken a series of molecular dynamic simulations, as suggested by R1, which do indeed strengthen our mechanistic interpretations. We discuss these new MD simulations at length in the Results section on page 10, lines 259-288, in two new figures (Fig. 6 and SI Fig. 7), and in the Discussion on page 12 lines 330-337. We use this new information to modify our proposed mechanism, specifically in relation to the N6 amine group and the N7, which is presented in Fig. 7. These new sections also respond to the concerns of other reviewers.

R1': I think there is a problem with the overall charges. In Fig. 7a ADP carries 3 negative charges, one on its alpha phosphate and two on its beta phosphate. In fact, one of the O- of the beta phosphate is the nucleophile that initiates ATP synthesis. However, in both the section about MD simulations and the legends of Figs. 6 and 7, ADP is described as having a 2- valence. Because the MD result is based on the electrostatic properties of the reactants one negative charge difference should have a major impact. This has to be clarified and revised. 

In Fig. 7b the ferric ion might be tetrahedral but, again, the coordination sphere is not well defined. It may be out of the scope of this paper, but better modeling would come out of a QM/MM analysis of the proposed structures.

R: We thank R1 again for their helpful comments and hope that we have addressed them to their satisfaction. The revisions very clearly improve the paper.

R1': You are welcome and please fix the remaining problems.

Reviewer #3:

The authors have made considerable, significant and positive changes to their original submission. It is clear from the combined reviewers comments that many different features of this paper have benefitted from these changes. 

From the perspective of this reviewer in particular, (i) the mechanistic aspects of ADP phosphorylation via AcP in the presence of Fe3+ has now been nuanced significantly, in line with accepted chemical precedence. (ii) Discussions around the "value" of mild, non-enzymatic ADP phosphorylation to a burgeoning prebiotic "organism" have also been modified and some additional caution introduced. 

The question as to how, where and when nucleotide phosphorylation became a fundamental component of primitive biological organisms is a foundational one in origin-of-life studies and its discussion is relevant to the current paper. However, the experimental offerings here are not focused on addressing that question, but rather on questions of chemistry. In this regard, I'm happy that the authors have sufficiently strengthened these chemical features, whilst modulating the language around potential "value" judgements which are less well substantiated. 

Reviewer #4:

The authors have taken into account prior recommendations and made changes as needed. Though there are different perspectives on this time of history (whether life was autotrophic early on is an open question), the authors have approached the subject reasonably.

---

## [Editor Report · Decision Letter 3]

30 Aug 2022

Dear Nick,

Thank you for the submission of your revised Research Article "A prebiotic basis for ATP as the universal energy currency" for publication in PLOS Biology. On behalf of my colleagues and the Academic Editor, Matthew Pasek, I am pleased to say that we can in principle accept your manuscript for publication, provided you address any remaining formatting and reporting issues. These will be detailed in an email you should receive within 2-3 business days from our colleagues in the journal operations team; no action is required from you until then. Please note that we will not be able to formally accept your manuscript and schedule it for publication until you have completed any requested changes.

Best wishes,

Roli

Senior Editor

PLOS Biology

rroberts@plos.org